# Algorithms with Logarithmic or Sublinear Regret for Constrained Contextual Bandits

**Huasen Wu**
University of California at Davis
hswu@ucdavis.edu

**R. Srikant**
University of Illinois at Urbana-Champaign
rsrikant@illinois.edu

**Xin Liu**
University of California at Davis
liu@cs.ucdavis.edu

**Chong Jiang**
University of Illinois at Urbana-Champaign
jiang17@illinois.edu

## Abstract

We study contextual bandits with budget and time constraints, referred to as *constrained contextual bandits*. The time and budget constraints significantly complicate the exploration and exploitation tradeoff because they introduce complex coupling among contexts over time. To gain insight, we first study unit-cost systems with known context distribution. When the expected rewards are known, we develop an approximation of the oracle, referred to Adaptive-Linear-Programming (ALP), which achieves near-optimality and only requires the ordering of expected rewards. With these highly desirable features, we then combine ALP with the upper-confidence-bound (UCB) method in the general case where the expected rewards are unknown *a priori*. We show that the proposed UCB-ALP algorithm achieves logarithmic regret except for certain boundary cases. Further, we design algorithms and obtain similar regret bounds for more general systems with unknown context distribution and heterogeneous costs. To the best of our knowledge, this is the first work that shows how to achieve logarithmic regret in constrained contextual bandits. Moreover, this work also sheds light on the study of computationally efficient algorithms for general constrained contextual bandits.

## 1 Introduction

The contextual bandit problem [1, 2, 3] is an important extension of the classic multi-armed bandit (MAB) problem [4], where the agent can observe a set of features, referred to as *context*, before making a decision. After the random arrival of a context, the agent chooses an action and receives a random reward with expectation depending on both the context and action. To maximize the total reward, the agent needs to make a careful tradeoff between taking the best action based on the historical performance (exploitation) and discovering the potentially better alternative actions under a given context (exploration). This model has attracted much attention as it fits the personalized service requirement in many applications such as clinical trials, online recommendation, and online hiring in crowdsourcing. Existing works try to reduce the regret of contextual bandits by leveraging the structure of the context-reward models such as linearity [5] or similarity [6], and more recent work [7] focuses on computationally efficient algorithms with minimum regret. For Markovian context arrivals, algorithms such as UCRL [8] for more general reinforcement learning problem can be used to achieve logarithmic regret.

However, traditional contextual bandit models do not capture an important characteristic of real systems: in addition to time, there is usually a cost associated with the resource consumed by each action and the total cost is limited by a budget in many applications. Taking crowdsourcing [9] as an example, the budget constraint for a given set of tasks will limit the number of workers that an employer can hire. Another example is the clinical trials [10], where each treatment is usually costly and the budget of a trial is limited. Although budget constraints have been studied in non-contextual bandits where logarithmic or sublinear regret is achieved [11, 12, 13, 14, 15, 16], as we will see later, these results are inapplicable in the case with observable contexts.

In this paper, we study contextual bandit problems with budget and time constraints, referred to as *constrained contextual bandits*, where the agent is given a budget $B$ and a time-horizon $T$. In addition to a reward, a cost is incurred whenever an action is taken under a context. The bandit process ends when the agent runs out of either budget or time. The objective of the agent is to maximize the expected total reward subject to the budget and time constraints. We are interested in the regime where $B$ and $T$ grow towards infinity proportionally.

The above constrained contextual bandit problem can be viewed as a special case of Resourceful Contextual Bandits (RCB) [17]. In [17], RCB is studied under more general settings with possibly infinite contexts, random costs, and multiple budget constraints. A Mixture_Elimination algorithm is proposed and shown to achieve $O(\sqrt{T})$ regret. However, the benchmark for the definition of regret in [17] is restricted to within a finite policy set. Moreover, the Mixture_Elimination algorithm suffers high complexity and the design of computationally efficient algorithms for such general settings is still an open problem.

To tackle this problem, motivated by certain applications, we restrict the set of parameters in our model as follows: we assume finite discrete contexts, fixed costs, and a single budget constraint. This simplified model is justified in many scenarios such as clinical trials [10] and rate selection in wireless networks [18]. More importantly, these simplifications allow us to design easily-implementable algorithms that achieve $O(\log T)$ regret (except for a set of parameters of zero Lebesgue measure, which we refer to as boundary cases), where the regret is defined more naturally as the performance gap between the proposed algorithm and *the oracle*, i.e., the optimal algorithm with known statistics.

Even with simplified assumptions considered in this paper, the exploration-exploitation tradeoff is still challenging due to the budget and time constraints. The key challenge comes from the complexity of the oracle algorithm. With budget and time constraints, the oracle algorithm cannot simply take the action that maximizes the instantaneous reward. In contrast, it needs to balance between the instantaneous and long-term rewards based on the current context and the remaining budget. In principle, dynamic programming (DP) can be used to obtain this balance. However, using DP in our scenario incurs difficulties in both algorithm design and analysis: first, the implementation of DP is computationally complex due to the curse of dimensionality; second, it is difficult to obtain a benchmark for regret analysis, since the DP algorithm is implemented in a recursive manner and its expected total reward is hard to be expressed in a closed form; third, it is difficult to extend the DP algorithm to the case with unknown statistics, due to the difficulty of evaluating the impact of estimation errors on the performance of DP-type algorithms.

To address these difficulties, we first study approximations of the oracle algorithm when the system statistics are known. Our key idea is to approximate the oracle algorithm with linear programming (LP) that relaxes the hard budget constraint to an average budget constraint. When fixing the average budget constraint at $B/T$, this LP approximation provides an upper bound on the expected total reward, which serves as a good benchmark in regret analysis. Further, we propose an Adaptive Linear Programming (ALP) algorithm that adjusts the budget constraint to the *average remaining budget* $b_\tau/\tau$, where $\tau$ is the remaining time and $b_\tau$ is the remaining budget. Note that although the idea of approximating a DP problem with an LP problem has been widely studied in literature (e.g., [17, 19]), the design and analysis of ALP here is quite different. In particular, we show that ALP achieves $O(1)$ regret, i.e., its expected total reward is within a constant independent of $T$ from the optimum, except for certain boundaries. This ALP approximation and its regret analysis make an important step towards achieving logarithmic regret for constrained contextual bandits.

Using the insights from the case with known statistics, we study algorithms for constrained contextual bandits with unknown expected rewards. Complicated interactions between information acquisition and decision making arise in this case. Fortunately, the ALP algorithm has a highly desirable property that it only requires the ordering of the expected rewards and can tolerate certain estimation errors of system parameters. This property allows us to combine ALP with estimation methods that can efficiently provide a correct rank of the expected rewards. In this paper, we propose a UCB-ALP algorithm by combining ALP with the upper-confidence-bound (UCB) method [4]. We show that UCB-ALP achieves $O(\log T)$ regret except for certain boundary cases, where its regret is $O(\sqrt{T})$. We note that UCB-type algorithms are proposed in [20] for non-contextual bandits with concave rewards and convex constraints, and further extended to linear contextual bandits. However, [20] focuses on static contexts[1] and achieves $O(\sqrt{T})$ regret in our setting since it uses a fixed budget constraint in each round. In comparison, we consider random context arrivals and use an adaptive

budget constraint to achieve logarithmic regret. To the best of our knowledge, this is the first work that shows how to achieve logarithmic regret in constrained contextual bandits. Moreover, the proposed UCB-ALP algorithm is quite computationally efficient and we believe these results shed light on addressing the open problem of general constrained contextual bandits.

Although the intuition behind ALP and UCB-ALP is natural, the rigorous analysis of their regret is non-trivial since we need to consider many interacting factors such as action/context ranking errors, remaining budget fluctuation, and randomness of context arrival. We evaluate the impact of these factors using a series of novel techniques, e.g., the method of showing concentration properties under adaptive algorithms and the method of bounding estimation errors under random contexts. For the ease of exposition, we study the ALP and UCB-ALP algorithms in unit-cost systems with known context distribution in Sections 3 and 4, respectively. Then we discuss the generalization to systems with unknown context distribution in Section 5 and with heterogeneous costs in Section 6, which are much more challenging and the details can be found in the supplementary material.

## 2 System Model

We consider a contextual bandit problem with a context set $\mathcal{X} = \{1, 2, \ldots, J\}$ and an action set $\mathcal{A} = \{1, 2, \ldots, K\}$. At each round $t$, a context $X_t$ arrives independently with identical distribution $\mathbb{P}\{X_t = j\} = \pi_j$, $j \in \mathcal{X}$, and each action $k \in \mathcal{A}$ generates a non-negative reward $Y_{k,t}$. Under a given context $X_t = j$, the reward $Y_{k,t}$'s are independent random variables in $[0, 1]$. The conditional expectation $\mathbb{E}[Y_{k,t}|X_t = j] = u_{j,k}$ is unknown to the agent. Moreover, a cost is incurred if action $k$ is taken under context $j$. To gain insight into constrained contextual bandits, we consider fixed and known costs in this paper, where the cost is $c_{j,k} > 0$ when action $k$ is taken under context $j$. Similar to traditional contextual bandits, the context $X_t$ is observable at the beginning of round $t$, while only the reward of the action taken by the agent is revealed at the end of round $t$.

At the beginning of round $t$, the agent observes the context $X_t$ and takes an action $A_t$ from $\{0\} \cup \mathcal{A}$, where "0" represents a *dummy* action that the agent skips the current context. Let $Y_t$ and $Z_t$ be the reward and cost for the agent in round $t$, respectively. If the agent takes an action $A_t = k > 0$, then the reward is $Y_t = Y_{k,t}$ and the cost is $Z_t = c_{X_t,k}$. Otherwise, if the agent takes the dummy action $A_t = 0$, neither reward nor cost is incurred, i.e., $Y_t = 0$ and $Z_t = 0$. In this paper, we focus on contextual bandits with a known time-horizon $T$ and limited budget $B$. The bandit process ends when the agent runs out of the budget or at the end of time $T$.

A contextual bandit algorithm $\Gamma$ is a function that maps the historical observations $\mathcal{H}_{t-1} = (X_1, A_1, Y_1; X_2, A_2, Y_2; \ldots; X_{t-1}, A_{t-1}, Y_{t-1})$ and the current context $X_t$ to an action $A_t \in \{0\} \cup \mathcal{A}$. The objective of the algorithm is to maximize the expected total reward $U_\Gamma(T, B)$ for a given time-horizon $T$ and a budget $B$, i.e.,

$$\text{maximize}_\Gamma \qquad U_\Gamma(T, B) = \mathbb{E}_\Gamma \Big[ \sum_{t=1}^{T} Y_t \Big]$$

$$\text{subject to} \qquad \sum_{t=1}^{T} Z_t \leq B,$$

where the expectation is taken over the distributions of contexts and rewards. Note that we consider a "hard" budget constraint, i.e., the total costs should not be greater than $B$ under any realization.

We measure the performance of the algorithm $\Gamma$ by comparing it with the oracle, which is the optimal algorithm with known statistics, including the knowledge of $\pi_j$'s, $u_{j,k}$'s, and $c_{j,k}$'s. Let $U^*(T, B)$ be the expected total reward obtained by the oracle algorithm. Then, the regret of the algorithm $\Gamma$ is defined as

$$R_\Gamma(T, B) = U^*(T, B) - U_\Gamma(T, B).$$

The objective of the algorithm is then to minimize the regret. We are interested in the asymptotic regime where the time-horizon $T$ and the budget $B$ grow to infinity proportionally, i.e., with a fixed ratio $\rho = B/T$.

## 3 Approximations of the Oracle

In this section, we study approximations of the oracle, where the statistics of bandits are known to the agent. This will provide a benchmark for the regret analysis and insights into the design of constrained contextual bandit algorithms.

As a starting point, we focus on unit-cost systems, i.e., $c_{j,k} = 1$ for each $j$ and $k$, from Section 3 to Section 6, which will be relaxed in Section 6. In unit-cost systems, the quality of action $k$ under context $j$ is fully captured by its expected reward $u_{j,k}$. Let $u_j^*$ be the highest expected reward under context $j$, and $k_j^*$ be the best action for context $j$, i.e., $u_j^* = \max_{k \in \mathcal{A}} u_{j,k}$ and $k_j^* = \arg\max_{k \in \mathcal{A}} u_{j,k}$. For ease of exposition, we assume that the best action under each context is unique, i.e., $u_{j,k} < u_j^*$ for all $j$ and $k \neq k_j^*$. Similarly, we also assume $u_1^* > u_2^* > \ldots > u_J^*$ for simplicity.

With the knowledge of $u_{j,k}$'s, the agent knows the best action $k_j^*$ and its expected reward $u_j^*$ under any context $j$. In each round $t$, the task of the oracle is deciding whether to take action $k_{X_t}^*$ or not depending on the remaining time $\tau = T - t + 1$ and the remaining budget $b_\tau$.

The special case of two-context systems ($J = 2$) is trivial, where the agent just needs to procrastinate for the better context (see Appendix D of the supplementary material). When considering more general cases with $J > 2$, however, it is computationally intractable to exactly characterize the oracle solution. Therefore, we resort to approximations based on linear programming (LP).

### 3.1  Upper Bound: Static Linear Programming

We propose an upper bound for the expected total reward $U^*(T, B)$ of the oracle by relaxing the hard constraint to an average constraint and solving the corresponding constrained LP problem. Specifically, let $p_j \in [0, 1]$ be the probability that the agent takes action $k_j^*$ for context $j$, and $1 - p_j$ be the probability that the agent skips context $j$ (i.e., taking action $A_t = 0$). Denote the probability vector as $\boldsymbol{p} = (p_1, p_2, \ldots, p_J)$. For a time-horizon $T$ and budget $B$, consider the following LP problem:

$$(\mathcal{LP}_{T,B}) \quad \text{maximize}_{\boldsymbol{p}} \quad \sum_{j=1}^{J} p_j \pi_j u_j^*, \tag{1}$$

$$\text{subject to} \quad \sum_{j=1}^{J} p_j \pi_j \leq B/T, \tag{2}$$

$$\boldsymbol{p} \in [0, 1]^J.$$

Define the following threshold as a function of the average budget $\rho = B/T$:

$$\tilde{j}(\rho) = \max\{j : \sum_{j'=1}^{j} \pi_{j'} \leq \rho\} \tag{3}$$

with the convention that $\tilde{j}(\rho) = 0$ if $\pi_1 > \rho$. We can verify that the following solution is optimal for $\mathcal{LP}_{T,B}$:

$$p_j(\rho) = \begin{cases} 1, & \text{if } 1 \leq j \leq \tilde{j}(\rho), \\ \frac{\rho - \sum_{j'=1}^{\tilde{j}(\rho)} \pi_{j'}}{\pi_{\tilde{j}(\rho)+1}}, & \text{if } j = \tilde{j}(\rho) + 1, \\ 0, & \text{if } j > \tilde{j}(\rho) + 1. \end{cases} \tag{4}$$

Correspondingly, the optimal value of $\mathcal{LP}_{T,B}$ is

$$v(\rho) = \sum_{j=1}^{\tilde{j}(\rho)} \pi_j u_j^* + p_{\tilde{j}(\rho)+1}(\rho) \pi_{\tilde{j}(\rho)+1} u_{\tilde{j}(\rho)+1}^*. \tag{5}$$

This optimal value $v(\rho)$ can be viewed as the maximum expected reward in a single round with average budget $\rho$. Summing over the entire horizon, the total expected reward becomes $\widehat{U}(T, B) = Tv(\rho)$, which is an upper bound of $U^*(T, B)$.

**Lemma 1.** *For a unit-cost system with known statistics, if the time-horizon is $T$ and the budget is $B$, then $\widehat{U}(T, B) \geq U^*(T, B)$.*

The proof of Lemma 1 is available in Appendix A of the supplementary material. With Lemma 1, we can bound the regret of any algorithm by comparing its performance with the upper bound $\widehat{U}(T, B)$ instead of $U^*(T, B)$. Since $\widehat{U}(T, B)$ has a simple expression, as we will see later, it significantly reduces the complexity of regret analysis.

## 3.2 Adaptive Linear Programming

Although the solution (4) provides an upper bound on the expected reward, using such a fixed algorithm will not achieve good performance as the ratio $b_\tau/\tau$, referred to as *average remaining budget*, fluctuates over time. We propose an Adaptive Linear Programming (ALP) algorithm that adjusts the threshold and randomization probability according to the instantaneous value of $b_\tau/\tau$.

Specifically, when the remaining time is $\tau$ and the remaining budget is $b_\tau = b$, we consider an LP problem $\mathcal{LP}_{\tau,b}$ which is the same as $\mathcal{LP}_{T,B}$ except that $B/T$ in Eq. (2) is replaced with $b/\tau$. Then, the optimal solution for $\mathcal{LP}_{\tau,b}$ can be obtained by replacing $\rho$ in Eqs. (3), (4), and (5) with $b/\tau$. The ALP algorithm then makes decisions based on this optimal solution.

**ALP Algorithm:** At each round $t$ with remaining budget $b_\tau = b$, obtain $p_j(b/\tau)$'s by solving $\mathcal{LP}_{\tau,b}$; take action $A_t = k^*_{X_t}$ with probability $p_{X_t}(b/\tau)$, and $A_t = 0$ with probability $1 - p_{X_t}(b/\tau)$.

The above ALP algorithm only requires the ordering of the expected rewards instead of their accurate values. This highly desirable feature allows us to combine ALP with classic MAB algorithms such as UCB [4] for the case without knowledge of expected rewards. Moreover, this simple ALP algorithm achieves very good performance within a constant distance from the optimum, i.e., $O(1)$ regret, except for certain boundary cases. Specifically, for $1 \leq j \leq J$, let $q_j$ be the cumulative probability defined as $q_j = \sum_{j'=1}^{j} \pi_{j'}$ with the convention that $q_0 = 0$. The following theorem states the near optimality of ALP.

**Theorem 1.** *Given any fixed $\rho \in (0,1)$, the regret of ALP satisfies:*
*1) (Non-boundary cases) if $\rho \neq q_j$ for any $j \in \{1, 2, \ldots, J-1\}$, then $R_{\mathrm{ALP}}(T, B) \leq \frac{u_1^* - u_J^*}{1 - e^{-2\delta^2}}$, where $\delta = \min\{\rho - q_{\tilde{j}(\rho)}, q_{\tilde{j}(\rho)+1} - \rho\}$.*
*2) (Boundary cases) if $\rho = q_j$ for some $j \in \{1, 2, \ldots, J-1\}$, then $R_{\mathrm{ALP}}(T, B) \leq \Theta^{(\mathrm{o})}\sqrt{T} + \frac{u_1^* - u_J^*}{1 - e^{-2(\delta')^2}}$, where $\Theta^{(\mathrm{o})} = 2(u_1^* - u_J^*)\sqrt{\rho(1-\rho)}$ and $\delta' = \min\{\rho - q_{\tilde{j}(\rho)-1}, q_{\tilde{j}(\rho)+1} - \rho\}$.*

Theorem 1 shows that ALP achieves $O(1)$ regret except for certain boundary cases, where it still achieves $O(\sqrt{T})$ regret. This implies that the regret due to the linear relaxation is negligible in most cases. Thus, when the expected rewards are unknown, we can achieve low regret, e.g., logarithmic regret, by combining ALP with appropriate information-acquisition mechanisms.

***Sketch of Proof:*** Although the ALP algorithm seems fairly intuitive, its regret analysis is non-trivial. The key to the proof is to analyze the evolution of the remaining budget $b_\tau$ by mapping ALP to "sampling without replacement". Specifically, from Eq. (4), we can verify that when the remaining time is $\tau$ and the remaining budget is $b_\tau = b$, the system consumes one unit of budget with probability $b/\tau$, and consumes nothing with probability $1 - b/\tau$. When considering the remaining budget, the ALP algorithm can be viewed as "sampling without replacement". Thus, we can show that $b_\tau$ follows the hypergeometric distribution [23] and has the following properties:

**Lemma 2.** *Under the ALP algorithm, the remaining budget $b_\tau$ satisfies:*
*1) The expectation and variance of $b_\tau$ are $\mathbb{E}[b_\tau] = \rho\tau$ and $\mathrm{Var}(b_\tau) = \frac{T-\tau}{T-1}\tau\rho(1-\rho)$, respectively.*
*2) For any positive number $\delta$ satisfying $0 < \delta < \min\{\rho, 1-\rho\}$, the tail distribution of $b_\tau$ satisfies*

$$\mathbb{P}\{b_\tau < (\rho - \delta)\tau\} \leq e^{-2\delta^2\tau} \ \ and \ \ \mathbb{P}\{b_\tau > (\rho + \delta)\tau\} \leq e^{-2\delta^2\tau}.$$

Then, we prove Theorem 1 based on Lemma 2. Note that the expected total reward under ALP is $U_{\mathrm{ALP}}(T, B) = \mathbb{E}\left[\sum_{\tau=1}^{T} v(b_\tau/\tau)\right]$, where $v(\cdot)$ is defined in (5) and the expectation is taken over the distribution of $b_\tau$. For the non-boundary cases, the single-round expected reward satisfies $\mathbb{E}[v(b_\tau/\tau)] = v(\rho)$ if the threshold $\tilde{j}(b_\tau/\tau) = \tilde{j}(\rho)$ for all possible $b_\tau$'s. The regret then is bounded by a constant because the probability of the event $\tilde{j}(b_\tau/\tau) \neq \tilde{j}(\rho)$ decays exponentially due to the concentration property of $b_\tau$. For the boundary cases, we show the conclusion by relating the regret with the variance of $b_\tau$. Please refer to Appendix B of the supplementary material for details.

## 4 UCB-ALP Algorithm for Constrained Contextual Bandits

Now we get back to the constrained contextual bandits, where the expected rewards are unknown to the agent. We assume the agent knows the context distribution as [17], which will be relaxed in Section 5. Thanks to the desirable properties of ALP, the maxim of "optimism under uncertainty"

[8] is still applicable and ALP can be extended to the bandit settings when combined with estimation policies that can quickly provide correct ranking with high probability. Here, combining ALP with the UCB method [4], we propose a UCB-ALP algorithm for constrained contextual bandits.

## 4.1 UCB: Notations and Property

Let $C_{j,k}(t)$ be the number of times that action $k \in \mathcal{A}$ has been taken under context $j$ up to round $t$. If $C_{j,k}(t-1) > 0$, let $\bar{u}_{j,k}(t)$ be the empirical reward of action $k$ under context $j$, i.e., $\bar{u}_{j,k}(t) = \frac{1}{C_{j,k}(t-1)} \sum_{t'=1}^{t-1} Y_{t'} \mathbb{1}(X_{t'} = j, A_{t'} = k)$, where $\mathbb{1}(\cdot)$ is the indicator function. We define the UCB of $u_{j,k}$ at $t$ as $\hat{u}_{j,k}(t) = \bar{u}_{j,k}(t) + \sqrt{\frac{\log t}{2C_{j,k}(t-1)}}$ for $C_{j,k}(t-1) > 0$, and $\hat{u}_{j,k}(t) = 1$ for $C_{j,k}(t-1) = 0$. Furthermore, we define the UCB of the maximum expected reward under context $j$ as $\hat{u}_j^*(t) = \max_{k \in \mathcal{A}} \hat{u}_{j,k}(t)$. As suggested in [24], we use a smaller coefficient in the exploration term $\sqrt{\frac{\log t}{2C_{j,k}(t-1)}}$ than the traditional UCB algorithm [4] to achieve better performance.

We present the following property of UCB that is important in regret analysis.

**Lemma 3.** *For two context-action pairs, $(j, k)$ and $(j', k')$, if $u_{j,k} < u_{j',k'}$, then for any $t \le T$,*

$$\mathbb{P}\{\hat{u}_{j,k}(t) \ge \hat{u}_{j',k'}(t) | C_{j,k}(t-1) \ge \ell_{j,k}\} \le 2t^{-1}, \tag{6}$$

*where $\ell_{j,k} = \frac{2 \log T}{(u_{j',k'} - u_{j,k})^2}$.*

Lemma 3 states that for two context-action pairs, the ordering of their expected rewards can be identified correctly with high probability, as long as the suboptimal pair has been executed for sufficient times (on the order of $O(\log T)$). This property has been widely applied in the analysis of UCB-based algorithms [4, 13], and its proof can be found in [13, 25] with a minor modification on the coefficients.

## 4.2 UCB-ALP Algorithm

We propose a UCB-based adaptive linear programming (UCB-ALP) algorithm, as shown in Algorithm 1. As indicated by the name, the UCB-ALP algorithm maintains UCB estimates of expected rewards for all context-action pairs and then implements the ALP algorithm based on these estimates. Note that the UCB estimates $\hat{u}_j^*(t)$'s may be non-decreasing in $j$. Thus, the solution of $\mathcal{LP}_{\tau,b}$ based on $\hat{u}_j^*(t)$ depends on the actual ordering of $\hat{u}_j^*(t)$'s and may be different from Eq. (4). We use $\hat{p}_j(\cdot)$ rather than $p_j(\cdot)$ to indicate this difference.

---
**Algorithm 1** UCB-ALP

    **Input:** Time-horizon $T$, budget $B$, and context distribution $\pi_j$'s;
    **Init:** $\tau = T$, $b = B$;
        $C_{j,k}(0) = 0$, $\bar{u}_{j,k}(0) = 0$, $\hat{u}_{j,k}(0) = 1$, $\forall j \in \mathcal{X}$ and $\forall k \in \mathcal{A}$; $\hat{u}_j^*(0) = 1, \forall j \in \mathcal{X}$;
    **for** $t = 1$ **to** $T$ **do**
        $k_j^*(t) \leftarrow \arg\max_k \hat{u}_{j,k}(t)$, $\forall j$;
        $\hat{u}_j^*(t) \leftarrow \hat{u}_{j,k_j^*(t)}^*(t)$;
        **if** $b > 0$ **then**
            Obtain the probabilities $\hat{p}_j(b/\tau)$'s by solving $\mathcal{LP}_{\tau,b}$ with $u_j^*$ replaced by $\hat{u}_j^*(t)$;
            Take action $k_{X_t}^*(t)$ with probability $\hat{p}_{X_t}(b/\tau)$;
        **end if**
        Update $\tau$, $b$, $C_{j,k}(t)$, $\bar{u}_{j,k}(t)$, and $\hat{u}_{j,k}(t)$.
    **end for**

---

## 4.3 Regret of UCB-ALP

We study the regret of UCB-ALP in this section. Due to space limitations, we only present a sketch of the analysis. Specific representations of the regret bounds and proof details can be found in the supplementary material.

Recall that $q_j = \sum_{j'=1}^{j} \pi_{j'}$ $(1 \le j \le J)$ are the boundaries defined in Section 3. We show that as the budget $B$ and the time-horizon $T$ grow to infinity in proportion, the proposed UCB-ALP algorithm achieves logarithmic regret except for the boundary cases.

**Theorem 2.** *Given $\pi_j$'s, $u_{j,k}$'s and a fixed $\rho \in (0,1)$, the regret of UCB-ALP satisfies:*
*1) (Non-boundary cases) if $\rho \neq q_j$ for any $j \in \{1, 2, \ldots, J-1\}$, then the regret of UCB-ALP is*
$R_{\text{UCB-ALP}}(T, B) = O(JK \log T)$.
*2) (Boundary cases) if $\rho = q_j$ for some $j \in \{1, 2, \ldots, J-1\}$, then the regret of UCB-ALP is*
$R_{\text{UCB-ALP}}(T, B) = O(\sqrt{T} + JK \log T)$.

Theorem 2 differs from Theorem 1 by an additional term $O(JK \log T)$. This term results from using UCB to learn the ordering of expected rewards. Under UCB, each of the $JK$ content-action pairs should be executed roughly $O(\log T)$ times to obtain the correct ordering. For the non-boundary cases, UCB-ALP is order-optimal because obtaining the correct action ranking under each context will result in $O(\log T)$ regret [26]. Note that our results do not contradict the lower bound in [17] because we consider discrete contexts and actions, and focus on instance-dependent regret. For the boundary cases, we keep both the $\sqrt{T}$ and $\log T$ terms because the constant in the $\log T$ term is typically much larger than that in the $\sqrt{T}$ term. Therefore, the $\log T$ term may dominate the regret particularly when the number of context-action pairs is large for medium $T$. It is still an open problem if one can achieve regret lower than $O(\sqrt{T})$ in these cases.

***Sketch of Proof:*** We bound the regret of UCB-ALP by comparing its performance with the benchmark $\widehat{U}(T, B)$. The analysis of this bound is challenging due to the close interactions among different sources of regret and the randomness of context arrivals. We first partition the regret according to the sources and then bound each part of regret, respectively.

**Step 1: Partition the regret.** By analyzing the implementation of UCB-ALP, we show that its regret is bounded as

$$R_{\text{UCB-ALP}}(T, B) \leq R_{\text{UCB-ALP}}^{(a)}(T, B) + R_{\text{UCB-ALP}}^{(c)}(T, B),$$

where the first part $R_{\text{UCB-ALP}}^{(a)}(T, B) = \sum_{j=1}^{J} \sum_{k \neq k_j^*} (u_j^* - u_{j,k}) \mathbb{E}[C_{j,k}(T)]$ is the regret from action ranking errors within a context, and the second part $R_{\text{UCB-ALP}}^{(c)}(T, B) = \sum_{\tau=1}^{T} \mathbb{E}[v(\rho) - \sum_{j=1}^{J} \hat{p}_j (b_\tau / \tau) \pi_j u_j^*]$ is the regret from the fluctuations of $b_\tau$ and context ranking errors.

**Step 2: Bound each part of regret.** For the first part, we can show that $R_{\text{UCB-ALP}}^{(a)}(T, B) = O(\log T)$ using similar techniques for traditional UCB methods [25]. The major challenge of regret analysis for UCB-ALP then lies in the evaluation of the second part $R_{\text{UCB-ALP}}^{(c)}(T, B)$.

We first verify that the evolution of $b_\tau$ under UCB-ALP is similar to that under ALP and Lemma 2 still holds under UCB-ALP. With respect to context ranking errors, we note that unlike classic UCB methods, not all context ranking errors contribute to the regret due to the threshold structure of ALP. Therefore, we carefully categorize the context ranking results based on their contributions. We briefly discuss the analysis for the non-boundary cases here. Recall that $\tilde{j}(\rho)$ is the threshold for the static LP problem $\mathcal{LP}_{T,B}$. We define the following events that capture all possible ranking results based on UCBs:

$$\mathcal{E}_{\text{rank},0}(t) = \{\forall j \leq \tilde{j}(\rho), \hat{u}_j^*(t) > \hat{u}_{\tilde{j}(\rho)+1}^*(t); \forall j > \tilde{j}(\rho) + 1, \hat{u}_j^*(t) < \hat{u}_{\tilde{j}(\rho)+1}^*(t)\},$$

$$\mathcal{E}_{\text{rank},1}(t) = \{\exists j \leq \tilde{j}(\rho), \hat{u}_j^*(t) \leq \hat{u}_{\tilde{j}(\rho)+1}^*(t); \forall j > \tilde{j}(\rho) + 1, \hat{u}_j^*(t) < \hat{u}_{\tilde{j}(\rho)+1}^*(t)\},$$

$$\mathcal{E}_{\text{rank},2}(t) = \{\exists j > \tilde{j}(\rho) + 1, \hat{u}_j^*(t) \geq \hat{u}_{\tilde{j}(\rho)+1}^*(t)\}.$$

The first event $\mathcal{E}_{\text{rank},0}(t)$ indicates a roughly correct context ranking, because under $\mathcal{E}_{\text{rank},0}(t)$ UCB-ALP obtains a correct solution for $\mathcal{LP}_{\tau,b_\tau}$ if $b_\tau / \tau \in [q_{\tilde{j}(\rho)}, q_{\tilde{j}(\rho)+1}]$. The last two events $\mathcal{E}_{\text{rank},s}(t)$, $s = 1, 2$, represent two types of context ranking errors: $\mathcal{E}_{\text{rank},1}(t)$ corresponds to "certain contexts with above-threshold reward having lower UCB", while $\mathcal{E}_{\text{rank},2}(t)$ corresponds to "certain contexts with below-threshold reward having higher UCB". Let $T^{(s)} = \sum_{t=1}^{T} \mathbb{1}(\mathcal{E}_{\text{rank},s}(t))$ for $0 \leq s \leq 2$. We can show that the expected number of context ranking errors satisfies $\mathbb{E}[T^{(s)}] = O(JK \log T)$, $s = 1, 2$, implying that $R_{\text{UCB-ALP}}^{(c)}(T, B) = O(JK \log T)$. Summarizing the two parts, we have $R_{\text{UCB-ALP}}(T, B) = O(JK \log T)$ for the non-boundary cases. The regret for the boundary cases can be bounded using similar arguments.

**Key Insights from UCB-ALP:** Constrained contextual bandits involve complicated interactions between information acquisition and decision making. UCB-ALP alleviates these interactions by

approximating the oracle with ALP for decision making. This approximation achieves near-optimal performance while tolerating certain estimation errors of system statistics, and thus enables the combination with estimation methods such as UCB in unknown statistics cases. Moreover, the adaptation property of UCB-ALP guarantees the concentration property of the system status, e.g., $b_\tau/\tau$. This allows us to separately study the impact of action or context ranking errors and conduct rigorous analysis of regret. These insights can be applied in algorithm design and analysis for constrained contextual bandits under more general settings.

## 5    Bandits with Unknown Context Distribution

When the context distribution is unknown, a reasonable heuristic is to replace the probability $\pi_j$ in ALP with its empirical estimate, i.e., $\hat{\pi}_j(t) = \frac{1}{t}\sum_{t'=1}^{t} \mathbb{1}(X_{t'} = j)$. We refer to this modified ALP algorithm as Empirical ALP (EALP), and its combination with UCB as UCB-EALP.

The empirical distribution provides a maximum likelihood estimate of the context distribution and the EALP and UCB-EALP algorithms achieve similar performance as ALP and UCB-ALP, respectively, as observed in numerical simulations. However, a rigorous analysis for EALP and UCB-EALP is much more challenging due to the dependency introduced by the empirical distribution. To tackle this issue, our rigorous analysis focuses on a truncated version of EALP where we stop updating the empirical distribution after a given round. Using the method of bounded averaged differences based on coupling argument, we obtain the concentration property of the average remaining budget $b_\tau/\tau$, and show that this truncated EALP algorithm achieves $O(1)$ regret except for the boundary cases. The regret of the corresponding UCB-based version can by bounded similarly as UCB-ALP.

## 6    Bandits with Heterogeneous Costs

The insights obtained from unit-cost systems can also be used to design algorithms for heterogeneous cost systems where the cost $c_{j,k}$ depends on $j$ and $k$. We generalize the ALP algorithm to approximate the oracle, and adjust it to the case with unknown expected rewards. For simplicity, we assume the context distribution is known here, while the empirical estimate can be used to replace the actual context distribution if it is unknown, as discussed in the previous section.

With heterogeneous costs, the quality of an action $k$ under a context $j$ is roughly captured by its *normalized expected reward*, defined as $\eta_{j,k} = u_{j,k}/c_{j,k}$. However, the agent cannot only focus on the "best" action, i.e., $k_j^* = \arg\max_{k\in\mathcal{A}} \eta_{j,k}$, for context $j$. This is because there may exist another action $k'$ such that $\eta_{j,k'} < \eta_{j,k_j^*}$, but $u_{j,k'} > u_{j,k_j^*}$ (and of course, $c_{j,k'} > c_{j,k_j^*}$). If the budget allocated to context $j$ is sufficient, then the agent may take action $k'$ to maximize the expected reward. Therefore, to approximate the oracle, the ALP algorithm in this case needs to solve an LP problem accounting for all context-action pairs with an additional constraint that only one action can be taken under each context. By investigating the structure of ALP in this case and the concentration of the remaining budget, we show that ALP achieves $O(1)$ regret in non-boundary cases, and $O(\sqrt{T})$ regret in boundary cases. Then, an $\epsilon$-First ALP algorithm is proposed for the unknown statistics case where an exploration stage is implemented first and then an exploitation stage is implemented according to ALP.

## 7    Conclusion

In this paper, we study computationally-efficient algorithms that achieve logarithmic or sublinear regret for constrained contextual bandits. Under simplified yet practical assumptions, we show that the close interactions between the information acquisition and decision making in constrained contextual bandits can be decoupled by adaptive linear relaxation. When the system statistics are known, the ALP approximation achieves near-optimal performance, while tolerating certain estimation errors of system parameters. When the expected rewards are unknown, the proposed UCB-ALP algorithm leverages the advantages of ALP and UCB, and achieves $O(\log T)$ regret except for certain boundary cases, where it achieves $O(\sqrt{T})$ regret. Our study provides an efficient approach of dealing with the challenges introduced by budget constraints and could potentially be extended to more general constrained contextual bandits.

**Acknowledgements:** This research was supported in part by NSF Grants CCF-1423542, CNS-1457060, CNS-1547461, and AFOSR MURI Grant FA 9550-10-1-0573.

## Footnotes

[1]After the online publication of our preliminary version, two recent papers [21, 22] extend their previous work [20] to the dynamic context case, where they focus on possibly infinite contexts and achieve $O(\sqrt{T})$ regret, and [21] restricts to a finite policy set as [17].

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
