[Supplementary Material]

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

[2]For the case with $c_{j,k_1} = c_{j,k_2}$ for some $j$ and $k_1 \neq k_2$ (and $u_{j,k_1} \neq u_{j,k_2}$), we can correctly remove the suboptimal action with high probability by comparing their empirical rewards $\bar{u}_{j,k_1} = \bar{u}_{j,k_2}$.

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

# Appendices

## A  Proof of Lemma 1: Upper Bound

We prove Lemma 1 by comparing $\widehat{U}(T, B)$ with the expected total reward under any feasible algorithm satisfying the budget constraint.

Let $C_j$ be the number of rounds that an action is taken under context $j$ for any realization under any feasible algorithm with known statistics. Let $p_j = \mathbb{E}[C_j]/(\pi_j T)$, which satisfies $0 \le p_j \le 1$. Then the expected total reward becomes $\sum_{j=1}^{J} u_j^* \mathbb{E}[C_j] = T \sum_{j=1}^{J} p_j \pi_j u_j^*$. Further, because the hard budget constraint is met for all realizations, i.e., $\sum_{j=1}^{J} C_j \le B$, we have $\sum_{j=1} p_j \pi_j = \sum_j E[C_j]/T \le B/T$. Thus, the expected total reward obtained by any feasible algorithm, including the oracle algorithm, is upper bounded by $\widehat{U}(T, B)$.

## B  Proof of Theorem 1: Near Optimality of ALP

### B.1  Proof of Lemma 2: Evolution of Remaining Budget

The evolution of the remaining budget $b_\tau$ is critical for evaluating the expected total reward under ALP. We prove Lemma 2 by casting ALP to a sampling problem without replacement.

From the implementation of ALP, we can verify that when the remaining time is $\tau$ and remaining budget is $b_\tau = b$, the system consumes one unit of budget with probability $b/\tau$, and consumes nothing with probability $1 - b/\tau$. Thus, when focusing on the remaining budget, we can view the ALP algorithm as a sampling problem without replacement as follows.

**Mapping ALP to Sampling without Replacement:** Consider $T$ balls in an urn, including $B$ black balls and $T - B$ white balls. Running ALP is equivalent to randomly drawing a ball without replacement. Taking an action $A_t > 0$ is equivalent to drawing a black ball and taking the dummy action $A_t = 0$ is equivalent to drawing a white ball. The event that $b_\tau = b$ is equivalent to the event that the agent draws $T - \tau$ balls, and the number of drawn black balls is $B - b$.

Based on the above mapping and using its symmetric property, we know that $b_\tau$ follows the hypergeometric distribution [23] and complete the proof of Lemma 2.

### B.2  Part 1: Non-Boundary Cases

According to Lemma 1, $\widehat{U}(T, B)$ is an upper bound on the total expected reward. Thus,

$$U^*(T, B) - U_{\text{ALP}}(T, B) \le \widehat{U}(T, B) - U_{\text{ALP}}(T, B) = \sum_{\tau=1}^{T} \left\{ v(\rho) - \mathbb{E}[v(b_\tau/\tau)] \right\}. \qquad (7)$$

To evaluate the gap between the single-round values, we define an auxiliary function $\tilde{v}(b/\tau)$ for a given $\rho$ as follows:

$$\tilde{v}(b/\tau) = \sum_{j=1}^{\tilde{j}(\rho)} \pi_j u_j^* + \pi_{\tilde{j}(\rho)+1} \tilde{p}_{\tilde{j}(\rho)+1}(b/\tau) u_{\tilde{j}(\rho)+1}^*, \qquad (8)$$

where

$$\tilde{p}_{\tilde{j}(\rho)+1}(b/\tau) = \frac{b/\tau - \sum_{j=1}^{\tilde{j}(\rho)} \pi_j}{\pi_{\tilde{j}(\rho)+1}}.$$

This auxiliary function bridges the gap of single-round values, $v(\rho)$ and $\mathbb{E}[v(b_\tau/\tau)]$, as follows:

First, we note that $\tilde{v}(b/\tau)$ uses the same threshold $\tilde{j}(\rho)$ as in $v(\rho)$. The only difference between $\tilde{v}(b/\tau)$ and $v(\rho)$ is that $\tilde{v}(b/\tau)$ uses the instantaneous average budget $b/\tau$ instead of the fixed average budget $\rho$. Considering all possible $b$'s and according to Lemma 2, we have

$$v(\rho) - \mathbb{E}[\tilde{v}(b_\tau/\tau)] = \left\{ \rho - \mathbb{E}[b_\tau/\tau] \right\} u_{\tilde{j}(\rho)+1}^* = 0. \qquad (9)$$

Second, compared with $v(b/\tau)$, the difference of the auxiliary function $\tilde{v}(b/\tau)$ comes from the event of $\tilde{j}(b/\tau) \neq \tilde{j}(\rho)$, which only occurs when $b/\tau < q_{\tilde{j}(\rho)}$ or $b/\tau > q_{\tilde{j}(\rho)+1}$. Because $\rho \neq q_j$, $1 \leq j \leq J-1$, there exists a positive number $\delta = \min\{\rho - q_{\tilde{j}(\rho)}, q_{\tilde{j}(\rho)+1} - \rho\}$ such that for all $\rho - \delta \leq \rho' < \rho + \delta$, the threshold under $\rho'$ is the same as that under $\rho$, i.e., $\tilde{j}(\rho') = \tilde{j}(\rho)$. Therefore, for all $b$ satisfying $\rho - \delta \leq b/\tau \leq \rho + \delta$, $v(b/\tau) = \tilde{v}(b/\tau)$.

If $b/\tau < \rho - \delta$, then

$$
\tilde{v}(b/\tau) - v(b/\tau)
$$
$$
= \left[ \sum_{j=\tilde{j}(b/\tau)+1}^{\tilde{j}(\rho)} \pi_j u_j^* + \left(\frac{b}{\tau} - q_{\tilde{j}(\rho)}\right) u_{\tilde{j}(\rho)+1}^* \right] - \left(\frac{b}{\tau} - q_{\tilde{j}(b/\tau)}\right) u_{\tilde{j}(b/\tau)+1}^*
$$
$$
\leq \left[ u_1^* \sum_{j=\tilde{j}(b/\tau)+1}^{\tilde{j}(\rho)} \pi_j + \left(\frac{b}{\tau} - q_{\tilde{j}(\rho)}\right) u_{\tilde{j}(\rho)+1}^* \right] - \left(\frac{b}{\tau} - q_{\tilde{j}(b/\tau)}\right) u_{\tilde{j}(\rho)+1}^*
$$
$$
\leq (u_1^* - u_{\tilde{j}(\rho)+1}^*) \sum_{j=\tilde{j}(b/\tau)+1}^{\tilde{j}(\rho)} \pi_j
$$
$$
\leq q_{\tilde{j}(\rho)}(u_1^* - u_J^*). \tag{10}
$$

Similarly, if $b/\tau > \rho + \delta$, then

$$
\tilde{v}(b/\tau) - v(b/\tau)
$$
$$
= \left(\frac{b}{\tau} - q_{\tilde{j}(\rho)}\right) u_{\tilde{j}(\rho)+1}^* - \left[ \sum_{j=\tilde{j}(\rho)+1}^{\tilde{j}(b/\tau)} \pi_j u_j^* + \left(\frac{b}{\tau} - q_{\tilde{j}(b/\tau)}\right) u_{\tilde{j}(b/\tau)+1}^* \right]
$$
$$
\leq (1 - q_{\tilde{j}(\rho)})(u_1^* - u_J^*). \tag{11}
$$

Summing all the above three cases ($\rho - \delta \leq b/\tau \leq \rho + \delta$, $b/\tau < \rho - \delta$, and $b/\tau > \rho + \delta$) and using Eq. (9), we have

$$
v(\rho) - \mathbb{E}[v(b_\tau/\tau)]
$$
$$
= v(\rho) - \mathbb{E}[\tilde{v}(b_\tau/\tau)] + \mathbb{E}[\tilde{v}(b_\tau/\tau) - v(b_\tau/\tau)]
$$
$$
= \sum_{b < \tau(\rho-\delta) \text{ or } b > \tau(\rho+\delta)} \mathbb{P}(b_\tau = b)[\tilde{v}(b/\tau) - v(b/\tau)]
$$
$$
\leq q_{\tilde{j}(\rho)}(u_1^* - u_J^*)\mathbb{P}\{b_\tau < \tau(\rho - \delta)\}
$$
$$
+ (1 - q_{\tilde{j}(\rho)})(u_1^* - u_J^*)\mathbb{P}\{b_\tau > \tau(\rho + \delta)\}
$$
$$
\leq (u_1^* - u_J^*)e^{-2\delta^2\tau}. \tag{12}
$$

Part 1 of Theorem 1 then follows by substituting Eq. (12) into Eq. (7).

### B.3 Part 2: Boundary Cases

The proof of Part 2 of Theorem 1 is similar to that of Part 1. Specifically, when $\rho = q_{\tilde{j}(\rho)}$, let $\delta' = \min\{\rho - q_{\tilde{j}(\rho)-1}, q_{\tilde{j}(\rho)+1} - \rho\}$. From the proof of Part 1, we know that

$$
v(\rho) - \mathbb{E}[\tilde{v}(b_\tau/\tau)] = 0. \tag{13}
$$

In addition, if $\rho \leq b/\tau \leq \rho + \delta'$, then $\tilde{j}(b/\tau) = \tilde{j}(\rho)$ and $v(b/\tau) = \tilde{v}(b/\tau)$.

If $\rho - \delta' \le b/\tau < \rho$, we have $\tilde{j}(b/\tau) = \tilde{j}(\rho) - 1$, and

$$
\begin{aligned}
&\tilde{v}(b/\tau) - v(b/\tau) \\
=\ & \pi_{\tilde{j}(\rho)} u^*_{\tilde{j}(\rho)} + (\frac{b}{\tau} - q_{\tilde{j}(\rho)}) u^*_{\tilde{j}(\rho)+1} - (\frac{b}{\tau} - q_{\tilde{j}(b/\tau)}) u^*_{\tilde{j}(b/\tau)+1} \\
=\ & (\pi_{\tilde{j}(\rho)} + q_{\tilde{j}(\rho)-1} - \frac{b}{\tau}) u^*_{\tilde{j}(\rho)} + (\frac{b}{\tau} - q_{\tilde{j}(\rho)}) u^*_{\tilde{j}(\rho)+1} \\
=\ & (\rho - \frac{b}{\tau}) u^*_{\tilde{j}(\rho)} + (\frac{b}{\tau} - \rho) u^*_{\tilde{j}(\rho)+1} \\
\le\ & |\frac{b}{\tau} - \rho| (u^*_1 - u^*_J).
\end{aligned}
\tag{14}
$$

Moreover, we still have (10) if $b/\tau < \rho - \delta'$, and (11) if $b/\tau > \rho + \delta'$.

Compared with the proof of Part 1, we know that the only difference relies on the case of $\rho - \delta' \le b/\tau < \rho$. Thus, summing all the above cases and using the results in the analysis of Part 1, we have

$$
\mathbb{E}[\tilde{v}(b_\tau/\tau) - v(b_\tau/\tau)] \le (u^*_1 - u^*_J)\{\mathbb{E}[|b_\tau/\tau - \rho|] + e^{-2(\delta')^2\tau}\} \le (u^*_1 - u^*_J)\Big[\sqrt{\frac{\mathrm{Var}(b_\tau)}{\tau^2}} + e^{-2(\delta')^2\tau}\Big].
$$

Consequently,

$$
\begin{aligned}
U^*(T,B) - U_{\mathrm{ALP}}(T,B) \ \le\ & \widehat{U}(T,B) - U_{\mathrm{ALP}}(T,B) \\
=\ & \sum_{\tau=1}^{T} \{v(\rho) - \mathbb{E}[v(b_\tau/\tau)]\} \\
=\ & \sum_{\tau=1}^{T} \{v(\rho) - \mathbb{E}[\tilde{v}(b_\tau/\tau)]\} + \sum_{\tau=1}^{T} \mathbb{E}[\tilde{v}(b_\tau/\tau) - v(b_\tau/\tau)] \\
\le\ & (u^*_1 - u^*_J) \sum_{\tau=1}^{T} \Big[\sqrt{\frac{\mathrm{Var}(b_\tau)}{\tau^2}} + e^{-2(\delta')^2\tau}\Big] \\
=\ & (u^*_1 - u^*_J) \sum_{\tau=1}^{T} \Big[\sqrt{\frac{(T-\tau)\rho(1-\rho)}{(T-1)\tau}} + e^{-2(\delta')^2\tau}\Big] \\
\le\ & (u^*_1 - u^*_J)\sqrt{\rho(1-\rho)} \sum_{\tau=1}^{T} \sqrt{\frac{1}{\tau}} + \frac{u^*_1 - u^*_J}{1 - e^{-2(\delta')^2}} \\
\le\ & 2\sqrt{\rho(1-\rho)}(u^*_1 - u^*_J)\sqrt{T} + \frac{u^*_1 - u^*_J}{1 - e^{-2(\delta')^2}}.
\end{aligned}
$$

## C  Proof of Theorem 2: Regret of UCB-ALP

We bound the regret of UCB-ALP by comparing its performance with the benchmark $\widehat{U}(T,B)$. To obtain this upper bound, we first partition the regret according to the sources and then bound each part of the regret, respectively.

Before presenting the proof, we first introduce a notation that will be widely used later. For contexts $j$ and $j'$, and an action $k$, let $\Delta^{(j')}_{j,k}$ be the difference between the expected reward for action $k$ under context $j$ and the highest expected reward under context $j'$, i.e., $\Delta^{(j')}_{j,k} = u^*_{j'} - u_{j,k}$. When $j' = j$, $\Delta^{(j)}_{j,k}$ is the difference of expected reward between the suboptimal action $k$ and the best action under context $j$.

## C.1 Step 1: Partition the Regret

Note that the total reward of the oracle solution $U^*(T, B) \leq \widehat{U}(T, B)$. Thus, we can bound the regret of UCB-ALP by comparing its total expected reward $U_{\text{UCB−ALP}}(T, B)$ with $\widehat{U}(T, B)$, i.e.,

$$
\begin{aligned}
& R_{\text{UCB−ALP}}(T, B) \\
= \ & U^*(T, B) - U_{\text{UCB−ALP}}(T, B) \\
\leq \ & \widehat{U}(T, B) - U_{\text{UCB−ALP}}(T, B) \\
= \ & Tv(\rho) - \sum_{j=1}^{J} \sum_{k=1}^{K} u_{j,k} \mathbb{E}[C_{j,k}(T)].
\end{aligned}
\tag{15}
$$

The total expected reward of UCB-ALP can be further divided as

$$
\begin{aligned}
& U_{\text{UCB−ALP}}(T, B) \\
= \ & \sum_{j=1}^{J} u_j^* \mathbb{E}\Big[\sum_{k=1}^{K} C_{j,k}(T)\Big] - \sum_{j=1}^{J} \sum_{k=1}^{K} \Delta_{j,k}^{(j)} \mathbb{E}[C_{j,k}(T)] \\
= \ & \sum_{j=1}^{J} u_j^* \mathbb{E}[C_j(T)] - \sum_{j=1}^{J} \sum_{k=1}^{K} \Delta_{j,k}^{(j)} \mathbb{E}[C_{j,k}(T)],
\end{aligned}
$$

where $C_j(T) = \sum_{k=1}^{K} C_{j,k}(T)$ is the total number that actions have been taken under context $j$ up to round $T$.

Consequently, the regret of UCB-ALP can be bounded as

$$
R_{\text{UCB−ALP}}(T, B) \leq R_{\text{UCB−ALP}}^{(a)}(T, B) + R_{\text{UCB−ALP}}^{(c)}(T, B),
\tag{16}
$$

where

$$
R_{\text{UCB−ALP}}^{(a)}(T, B) = \sum_{j=1}^{J} \sum_{k=1}^{K} \Delta_{j,k}^{(j)} \mathbb{E}[C_{j,k}(T)],
$$

and

$$
R_{\text{UCB−ALP}}^{(c)}(T, B) = \sum_{\tau=1}^{T} \mathbb{E}\Big[v(\rho) - \sum_{j=1}^{J} \hat{p}_j(b_\tau/\tau) \pi_j u_j^*\Big].
$$

Eq. (16) clearly shows that the regret of the UCB-ALP algorithm can be divided into two parts: the first part $R_{\text{UCB−ALP}}^{(a)}(T, B)$ is from taking suboptimal actions under a given context; the second part $R_{\text{UCB−ALP}}^{(c)}(T, B)$ is from the deviation of remaining budget $b_\tau$ and context ranking errors.

## C.2 Step 2: Bound Each Part of Regret

### C.2.1 Step 2.1: Bound of $R_{\textbf{UCB-ALP}}^{(a)}(T, B)$

For the regret from action ranking errors, we show in Lemma 4 that $R_{\text{UCB-ALP}}^{(a)}(T, B) = O(\log T)$ using similar techniques for traditional UCB methods [25].

**Lemma 4.** *Under UCB-ALP, the regret due to the action ranking errors within context $j$ satisfies*

$$
\begin{aligned}
& R_{UCB\text{-}ALP}^{(a)}(T, B) \\
\leq \ & \sum_{j=1}^{J} \sum_{k \neq k_j^*} \left[ \Big(\frac{2}{\Delta_{j,k}^{(j)}} + 2\Delta_{j,k}^{(j)}\Big) \log T + 2\Delta_{j,k}^{(j)} \right].
\end{aligned}
\tag{17}
$$

*Proof.* For $k \neq k_j^*$, let $\ell_{j,k}^{(j)} = \frac{2 \log T}{(\Delta_{j,k}^{(j)})^2}$. According to Lemma 3, we have

$$\mathbb{E}[C_{j,k}(T)]$$

$$\leq \quad \ell_{j,k}^{(j)} + \sum_{t=1}^{T} \mathbb{P}\{X_t = j, A_t = k, C_{j,k}(t-1) \geq \ell_{j,k}^{(j)}\}$$

$$\leq \quad \ell_{j,k}^{(j)} + \sum_{t=1}^{T} \mathbb{P}\{X_t = j, A_t = k | C_{j,k}(t-1) \geq \ell_{j,k}^{(j)}\}$$

$$\leq \quad \ell_{j,k}^{(j)} + \sum_{t=1}^{T} 2t^{-1}.$$

The conclusion then follows by the facts that $\sum_{t=1}^{T} t^{-1} \leq 1 + \log T$ and $R_{\text{UCB-ALP}}^{(a)}(T, B) = \sum_{j=1}^{J} \sum_{k \neq k_j^*} \Delta_{j,k}^{(j)} \mathbb{E}[C_{j,k}(T)]$. $\qquad \square$

### C.2.2 Step 2.2: Bound of $R_{\text{UCB-ALP}}^{(c)}(T, B)$

Next, we show that the second part $R_{\text{UCB-ALP}}^{(c)}(T, B) = O(\log T)$. **We first present the proof for the non-boundary cases**, and discuss the boundary cases later.

Note that we have separately considered the regret due to action ranking errors in $R_{\text{UCB-ALP}}^{(a)}(T, B)$ and we only need to consider the best action of each context for $R_{\text{UCB-ALP}}^{(c)}(T, B)$. Thus, we define $v_{\text{UCB-ALP}}^*(\tau, b_\tau)$ as follows:

$$v_{\text{UCB-ALP}}^*(\tau, b_\tau) = \sum_{j=1}^{J} \tilde{p}_j(b_\tau/\tau) \pi_j u_j^*.$$

Let $\Delta v_\tau$ be the single-round difference between UCB-ALP and the upper bound, i.e.,

$$\Delta v_\tau = v(\rho) - v_{\text{UCB-ALP}}^*(\tau, b_\tau).$$

Then $R_{\text{UCB-ALP}}^{(c)}(T, B) = \sum_{\tau=T}^{1} \mathbb{E}[\Delta v_\tau]$. We study the expectation $\mathbb{E}[\Delta v_\tau]$ under all possible situations. For a random variable $X$ and event $\mathcal{E}$, let $\mathbb{E}[X, \mathcal{E}] = \mathbb{E}[X \mathbb{1}(\mathcal{E})]$. Then, the expectation $\mathbb{E}[X] = \mathbb{E}[X, \mathcal{E}] + \mathbb{E}[X, \neg \mathcal{E}]$. Therefore,

$$\mathbb{E}[\Delta v_\tau] = \sum_{s=0}^{2} \mathbb{E}[\Delta v_\tau, \mathcal{E}_{\text{rank},s}(T - \tau + 1)]. \tag{18}$$

We first consider the case of $s = 0$ and convert the expectation value into other two cases. Considering all possible value of $b_\tau$, we have

$$\mathbb{E}[\Delta v_\tau, \mathcal{E}_{\text{rank},0}(T - \tau + 1)]$$

$$= \quad \sum_{b=0}^{B} \mathbb{E}[\Delta v_\tau | b_\tau = b, \mathcal{E}_{\text{rank},0}(T - \tau + 1)] \mathbb{P}\{b_\tau = b, \mathcal{E}_{\text{rank},0}(T - \tau + 1)\}. \tag{19}$$

For the probability, we have

$$\mathbb{P}\{b_\tau = b, \mathcal{E}_{\text{rank},0}(T - \tau + 1)\} = \mathbb{P}\{b_\tau = b\} - \sum_{s=1}^{2} \mathbb{P}\{b_\tau = b, \mathcal{E}_{\text{rank},s}(T - \tau + 1)\}. \tag{20}$$

For the conditioned expectation, we note that $\mathcal{E}_{\text{rank},0}(T - \tau + 1)$ provides a roughly correct context rank in the sense that if $b_\tau/\tau$ is close to $\rho$, then $v_{\text{UCB-ALP}}^*(\tau, b_\tau) = v(b_\tau/\tau)$, where $v(b_\tau/\tau)$ is the single round value with the correct context rank. Specifically, letting $\delta = \frac{1}{2} \min\{\rho - q_{\tilde{j}(\rho)}, q_{\tilde{j}(\rho)+1} - \rho\}$. If $b \in [\rho - \delta, \rho + \delta]$, then $v_{\text{UCB-ALP}}^*(\tau, b) = v(b/\tau)$, and thus,

$$\mathbb{E}[\Delta v_\tau | b_\tau = b, \mathcal{E}_{\text{rank},0}(T - \tau + 1)] = v(\rho) - v(b/\tau). \tag{21}$$

Combining Eqs. (19) $\sim$ (21) and using the facts that $v(\rho) \geq 0$ and $v^*_{\text{UCB-ALP}}(\tau, b) \geq 0$, we have

$$\mathbb{E}[\Delta v_\tau, \mathcal{E}_{\text{rank},0}(T - \tau + 1)]$$

$$\leq v(\rho) - \sum_{b=0}^{B} v(b/\tau)\mathbb{P}\{b_\tau = b\}$$

$$+ \sum_{s=1}^{2} \sum_{b=0}^{B} v(b/\tau)\mathbb{P}\{b_\tau = b, \mathcal{E}_{\text{rank},s}(T - \tau + 1)\}$$

$$+ \sum_{b \notin [\rho-\delta, \rho+\delta]} v(b/\tau)\mathbb{P}\{b_\tau = b, \mathcal{E}_{\text{rank},0}(T - \tau + 1)\}. \tag{22}$$

Recall that under UCB-ALP, the remaining budget $b_\tau$ follows the hypergeometric distribution. Using the same method as the analysis of Eq. (12), we have

$$v(\rho) - \sum_{b=0}^{B} v(b/\tau)\mathbb{P}\{b_\tau = b\} \leq (u_1^* - u_J^*)e^{-2\delta^2\tau}. \tag{23}$$

In addition,

$$\sum_{b \notin [\rho-\delta, \rho+\delta]} v(b/\tau)\mathbb{P}\{b_\tau = b, \mathcal{E}_{\text{rank},0}(T - \tau + 1)\} \leq \bar{u}^* \sum_{b \notin [\rho-\delta, \rho+\delta]} \mathbb{P}\{b_\tau = b\} \leq 2\bar{u}^* e^{-2\delta^2\tau}, \tag{24}$$

where $\bar{u}^* = \sum_{j=1}^{J} \pi_j u_j^*$ is the expected reward without budget constraint.

Moreover,

$$\sum_{b=0}^{B} v(b/\tau)\mathbb{P}\{b_\tau = b, \mathcal{E}_{\text{rank},s}(T - \tau + 1)\} \leq \bar{u}^* \mathbb{P}\{\mathcal{E}_{\text{rank},s}(T - \tau + 1)\}, \tag{25}$$

Substituting Eqs. (23) $\sim$ (25) into Eq. (22), we have

$$\mathbb{E}[\Delta v_\tau, \mathcal{E}_{\text{rank},0}(T - \tau + 1)] \leq [u_1^* - u_J^* + 2\bar{u}^*]e^{-2\delta^2\tau} + \bar{u}^* \sum_{s=1}^{2} \mathbb{P}\{\mathcal{E}_{\text{rank},s}(T - \tau + 1)\}. \tag{26}$$

When the rank is wrong, i.e., $1 \leq s \leq 2$, since $\Delta v_\tau \leq v(\rho)$ under any possible ranking results, we have

$$\mathbb{E}[\Delta v_\tau, \mathcal{E}_{\text{rank},s}(T - \tau + 1)] \leq v(\rho)\mathbb{P}\{\mathcal{E}_{\text{rank},s}(T - \tau + 1)\}. \tag{27}$$

Substituting Eqs. (26) and (27) into Eq. (18), we have

$$\mathbb{E}[\Delta v_\tau] \leq [u_1^* - u_J^* + 2\bar{u}^*]e^{-2\delta^2\tau} + [\bar{u}^* + v(\rho)] \sum_{s=1}^{2} \mathbb{P}\{\mathcal{E}_{\text{rank},s}(T - \tau + 1)\}.$$

Note that $R^{(c)}_{\text{UCB-ALP}}(T, B) = \sum_{\tau=1}^{T} \mathbb{E}[\Delta v_\tau]$. Thus

$$R^{(c)}_{\text{UCB-ALP}}(T, B) \leq \frac{[u_1^* - u_J^* + 2\bar{u}^*]e^{-2\delta^2}}{1 - e^{-2\delta^2}} + [\bar{u}^* + v(\rho)] \sum_{s=1}^{2} \mathbb{E}[T^{(s)}], \tag{28}$$

where $T^{(s)} = \sum_{t=1}^{T} \mathbb{1}(\mathcal{E}_{\text{rank},s}(t))$ ($s = 1, 2$) is the number of type-$s$ ranking errors.

Next, we bound the expected number of context ranking errors. From Lemma 3, we know that to obtain the correct ordering of two context-action pairs with high probability, the agent needs to execute the suboptimal context-action pair for enough times. Unlike traditional MABs, however, the context-action pair with the higher UCB in a round might not be executable, as the context of that round could be different. Fortunately, the following lemma will show that if the condition that causes the an context-action pair to be executed with a positive probability appears many times, the context-action pair will indeed be executed proportionally with high probability.

**Lemma 5.** *Assume $\mathcal{E}_t$'s, $\hat{\mathcal{E}}_t$'s are events in round $t$ ($1 \le t \le T$), satisfying $\mathbb{P}\{\mathcal{E}_t|\hat{\mathcal{E}}_t, \mathcal{H}_{1:t-1}\} = \mathbb{P}\{\mathcal{E}_t|\hat{\mathcal{E}}_t\} \ge p > 0$, where $\mathcal{H}_{1:t-1}$ is the filtration from 1 to $t-1$. Let $C(T) = \sum_{t=1}^{T} \mathbb{1}(\mathcal{E}_t)$ and $\hat{C}(T) = \sum_{t=1}^{T} \mathbb{1}(\hat{\mathcal{E}}_t)$. Then,*

$$\mathbb{P}\{C(T) \le (p-\epsilon)N, \hat{C}(T) \ge N\} \le e^{-2\epsilon^2 N}.$$

*Proof.* One may think the proof of this lemma is trivial because $\mathbb{P}\{C(T) \le (p-\epsilon)N, \hat{C}(T) \ge N\} \le \mathbb{P}\{C(T) \le (p-\epsilon)N|\hat{C}(T) \ge N\}$ and we can bound the right-hand-side using Chernoff bound. However, this is incorrect because although $\mathcal{E}_t$ is independent of the history given $\hat{\mathcal{E}}_t$, the event $\hat{\mathcal{E}}_{t+1}$ may depend on $\mathcal{E}(t)$.

We prove this lemma using the coupling argument. Let $S_t = \mathbb{1}(\mathcal{E}_t \cap \hat{\mathcal{E}}_t)$, and $C_S(T) = \sum_{t=1}^{T} S_t$. Then, we have

$$\mathbb{P}\{C(T) \le (p-\epsilon)N, \hat{C}(T) \ge N\} \le \mathbb{P}\{C_S(T) \le (p-\epsilon)N, \hat{C}(T) \ge N\}. \tag{29}$$

Now, we show $\mathbb{P}\{C_S(T) \le (p-\epsilon)N, \hat{C}(T) \ge N\} \le e^{-2\epsilon^2 N}$ using the coupling argument.

First, generate $W_1, W_2, \ldots, W_T$ i.i.d. according to Bernoulli distribution with $\mathbb{P}\{W_t = 1\} = p$.

Next, generate a sequences $(V_t', S_t', 1 \le t \le T)$ as follows:

For each $t$, generate $V_t'$ according to Bernoulli distribution with $\mathbb{P}\{V_t' = 1\} = \mathbb{P}\{\hat{\mathcal{E}}_t = 1|\mathbb{1}(\hat{\mathcal{E}}_{t'}) = V_{t'}', \mathbb{1}(\mathcal{E}_{t'}) = S_{t'}', 1 \le t' \le t-1\}$. Further, we generate $S_t'$ conditioned on the value of $V_t'$ and $W_t$. Specifically, let $C_{V'}(t) = \sum_{t'=1}^{t} V_t'$.

**1)** If $V_t' = 1$, generate $S_t'$ conditioned on $W_{C_{V'}(t)}$:
*a.* If $W_{C_{V'}(t)} = 1$, then $S_t' = 1$;
*b.* If $W_{C_{V'}(t)} = 0$, then generate $S_t'$ according to Bernoulli distribution with

$$\mathbb{P}\{S_t' = 1|W_{C_{V'}(t)} = 0\} = \frac{\mathbb{P}[\mathbb{1}(\mathcal{E}_t) = 1|\mathbb{1}(\hat{\mathcal{E}}_t) = 1] - p}{1 - p}.$$

**2)** If $V_t' = 0$, let $S_t' = 0$.

We can verify that $(V_t', S_t', 1 \le t \le T)$ has the same distribution as $(\mathbb{1}(\hat{\mathcal{E}}_t), S_t, 1 \le t \le T)$. Hence, $\mathbb{P}\{C_S(T) \le (p-\epsilon)N, \hat{C}(T) \ge N\} = \mathbb{P}\{\sum_{t=1}^{T} S_t' \le (p-\epsilon)N, \sum_{t=1}^{T} V_t' \ge N\}$.

On the other hand, from the generation of $S_t'$, we have $\sum_{t=1}^{T} S_t' \ge \sum_{t=1}^{C_{V'}(T)} W_t$. Thus, the event $\{\sum_{t=1}^{T} S_t' \le (p-\epsilon)N, \sum_{t=1}^{T} V_t' \ge N\}$ implies $\{\sum_{t=1}^{N} W_t \le (p-\epsilon)N\}$, and

$$\mathbb{P}\{\sum_{t=1}^{T} S_t' \le (p-\epsilon)N, \sum_{t=1}^{T} V_t' \ge N\} \le \mathbb{P}\{\sum_{t=1}^{N} W_t \le (p-\epsilon)N\} \le e^{-2\epsilon^2 N}. \tag{30}$$

The conclusion of the lemma then follows. $\square$

The following lemma bounds the expected number of context ranking errors.

**Lemma 6.** *Given $\pi_j$'s, $u_{j,k}$'s and a fixed $\rho \in (0,1)$, $\rho \ne q_j$ ($1 \le j \le J-1$), under the UCB-ALP algorithm, we have*

$$\mathbb{E}[T^{(1)}] \le \sum_{j=1}^{\tilde{j}(\rho)} \sum_{k=1}^{K} \frac{27 \log T}{2g_{\tilde{j}(\rho)+1}[\Delta_{\tilde{j}(\rho)+1,k}^{(j)}]^2} + 2K\tilde{j}(\rho) \log T + O(1),$$

$$\mathbb{E}[T^{(2)}] \le \sum_{j=\tilde{j}(\rho)+2}^{J} \sum_{k=1}^{K} \frac{27 \log T}{2g_j[\Delta_{j,k}^{(\tilde{j}(\rho)+1)}]^2} + 2K[J - \tilde{j}(\rho) - 1] \log T + O(1),$$

*where $g_j = \min\left\{\pi_j, \frac{1}{2}(\rho - q_{\tilde{j}(\rho)}), \frac{1}{2}(q_{\tilde{j}(\rho)+1} - \rho)\right\}$.*

*Proof.* We only prove the conclusion for the case of $s = 1$ as the other case can be analyzed similarly. From Algorithm 1, we can see that the evolution of the remaining budget also affects the execution of the UCB-ALP algorithm. Under the assumption of known context distribution, it can be verified that Lemma 2 holds under UCB-ALP, i.e., the remaining budget $b_\tau$ follows the hypergeometric distribution and has the properties described in Lemma 2. We define an event $\mathcal{E}_{\text{budget},0}(t)$ as follows,

$$\mathcal{E}_{\text{budget},0}(t) = \{(\rho - \delta)\tau \leq b_\tau \leq (\rho + \delta)\tau\},$$

where $\delta$ is given by

$$\delta = \frac{1}{2}\min\{\rho - q_{\tilde{j}(\rho)}, q_{\tilde{j}(\rho)+1} - \rho\}.$$

According to Lemma 2, we have

$$\mathbb{P}\{\neg\mathcal{E}_{\text{budget},0}(t)\} = \mathbb{P}\{b_\tau < (\rho - \delta)\tau\} + \mathbb{P}\{b_\tau > (\rho + \delta)\tau\} \leq 2e^{-2\delta^2(T-t+1)}.$$

Back to the ranking event $\mathcal{E}_{\text{rank},1}(t)$, we have

$$\mathbb{P}(\mathcal{E}_{\text{rank},1}(t)) \leq \mathbb{P}(\neg\mathcal{E}_{\text{budget},0}(t)) + \mathbb{P}(\mathcal{E}_{\text{rank},1}(t) \cap \mathcal{E}_{\text{budget},0}(t)).$$

Note that the event $\mathcal{E}_{\text{rank},1}(t)$ can be divided as follow:

$$\mathcal{E}_{\text{rank},1}(t) \subseteq \bigcup_{1 \leq j \leq \tilde{j}(\rho), 1 \leq k \leq K} \mathcal{E}_{\text{rank},1}^{(j,k)}(t),$$

where for $1 \leq j \leq \tilde{j}(\rho)$ and $1 \leq k \leq K$,

$$\mathcal{E}_{\text{rank},1}^{(j,k)}(t) = \{\forall j' > \tilde{j}(\rho) + 1, \hat{u}_{j'}^*(t) < \hat{u}_{\tilde{j}(\rho)+1}^*(t); \hat{u}_j^*(t) \leq \hat{u}_{\tilde{j}(\rho)+1}^*(t), k_j^*(t) = k\}.$$

Thus,

$$\mathbb{E}[T^{(1)}] = \sum_{t=1}^{T} \mathbb{P}(\mathcal{E}_{\text{rank},1}(t)) \leq \frac{2e^{-2\delta^2}}{1 - e^{-2\delta^2}} + \sum_{j=1}^{\tilde{j}(\rho)} \sum_{k=1}^{K} \mathbb{E}[N_{j,k}^{(1)}(T)],$$

$$(31)$$

where for $1 \leq j \leq \tilde{j}(\rho)$ and $1 \leq k \leq K$,

$$N_{j,k}^{(1)}(t) = \sum_{t'=1}^{t} \mathbb{1}(\mathcal{E}_{\text{rank},1}^{(j,k)}(t'), \mathcal{E}_{\text{budget},0}(t')).$$

Let $\hat{\ell}_{j_2,k}^{(j_1)} = \frac{2\log T}{g_{j_2}(1-\epsilon)^2(\Delta_{j_2,k}^{(j_1)})^2}$, where $g_{j_2} = \min\{\pi_{j_2}, \delta\}$ and $\epsilon \in (0, 1)$. Similar to the analysis of UCB in [4], we have

$$\mathbb{E}[N_{j,k}^{(1)}(T)] \leq \hat{\ell}_{\tilde{j}(\rho)-1,k}^{(j)} + \sum_{t=1}^{T} \mathbb{P}\{\mathcal{E}_{\text{rank},1}^{(j,k)}(t), \mathcal{E}_{\text{budget},0}(t), N_{j,k}^{(1)}(t-1) \geq \hat{\ell}_{\tilde{j}(\rho)+1,k}^{(j)}\}. \quad (32)$$

For each $t$ in the second term, we have

$$\mathbb{P}\{\mathcal{E}_{\text{rank},1}^{(j,k)}(t), \mathcal{E}_{\text{budget},0}(t), N_{j,k}^{(1)}(t-1) \geq \hat{\ell}_{\tilde{j}(\rho)+1,k}^{(j)}\}$$

$$\leq \mathbb{P}\{\mathcal{E}_{\text{rank},1}^{(j,k)}(t), \mathcal{E}_{\text{budget},0}(t)|C_{\tilde{j}(\rho)+1,k}(t-1) \geq g_{\tilde{j}(\rho)+1}(1-\epsilon)\hat{\ell}_{\tilde{j}(\rho)+1,k}^{(j)}\}$$

$$+ \mathbb{P}\{C_{\tilde{j}(\rho)+1,k}(t-1) < g_{\tilde{j}(\rho)+1}(1-\epsilon)\hat{\ell}_{\tilde{j}(\rho)+1,k}^{(j)}, N_{j,k}^{(1)}(t-1) \geq \hat{\ell}_{\tilde{j}(\rho)+1,k}^{(j)}\},$$

where $C_{\tilde{j}(\rho)+1,k}(t) = \sum_{t'=1}^{t} \mathbb{1}(X_{t'} = \tilde{j}(\rho) + 1, A_{t'} = k)$ is the number that the context-action pair $(\tilde{j}(\rho) + 1, k)$ has been executed up to round $t$.

For the first term, we note that the event $\{\mathcal{E}_{\mathrm{rank},1}^{(j,k)}(t), \mathcal{E}_{\mathrm{budget},0}(t)\}$ implies that $\hat{u}_{j,k_j^*}(t) \leq \hat{u}_{\tilde{j}(\rho)+1,k}(t)$. Because $u_{j,k_j^*} > u_{\tilde{j}(\rho)+1,k}$ for all $j \leq \tilde{j}(\rho)$ and $k$, according to Lemma 3, we have

$$\mathbb{P}\{\mathcal{E}_{\mathrm{rank},1}^{(j,k)}(t), \mathcal{E}_{\mathrm{budget},0}(t) | C_{\tilde{j}(\rho)+1,k}(t-1) \geq g_{\tilde{j}(\rho)+1}(1-\epsilon)\hat{\ell}_{\tilde{j}(\rho)+1,k}^{(j)}\}$$

$$\leq \mathbb{P}\{\hat{u}_{j,k_j^*}(t) \leq \hat{u}_{\tilde{j}(\rho)+1,k}(t) | C_{\tilde{j}(\rho)+1,k}(t-1) \geq g_{\tilde{j}(\rho)+1}(1-\epsilon)\hat{\ell}_{\tilde{j}(\rho)+1,k}^{(j)}\}$$

$$\leq 2t^{-1}. \tag{33}$$

For the second term, we note that since context $\tilde{j}(\rho)+1$ arrives with probability $\pi_{\tilde{j}(\rho)+1}$ independent of the observations, we have

$$\mathbb{P}\{X_t = \tilde{j}(\rho)+1, A_t = k | \mathcal{E}_{\mathrm{rank},1}^{(j,k)}(t), \mathcal{E}_{\mathrm{budget},0}(t)\} = \min\{\delta, \pi_{\tilde{j}(\rho)+1}\} = g_{\tilde{j}(\rho)+1}.$$

Thus, according to Lemma 5, we have

$$\mathbb{P}\{C_{\tilde{j}(\rho)+1,k}(t-1) < g_{\tilde{j}(\rho)+1}(1-\epsilon)\hat{\ell}_{\tilde{j}(\rho)+1,k}^{(j)}, N_{j,k}^{(1)}(t-1) \geq \hat{\ell}_{\tilde{j}(\rho)+1,k}^{(j)}\} \leq e^{-2\epsilon^2 \hat{\ell}_{\tilde{j}(\rho)+1,k}^{(j)}} \leq T^{-4}. \tag{34}$$

Substituting Eqs. (33) and (34) into Eq. (32), we have

$$\mathbb{E}[N_{j,k}^{(1)}(T)] \leq \hat{\ell}_{\tilde{j}(\rho)+1,k}^{(j)} + \sum_{t=1}^{T}(2t^{-1} + T^{-4}) \leq \hat{\ell}_{\tilde{j}(\rho)+1,k}^{(j)} + 2\log T + 2 + T^{-3}. \tag{35}$$

Substituting Eq. (35) to Eq. (31) and letting $\epsilon = 2/3$ in $\hat{\ell}_{\tilde{j}(\rho)-1,k}^{(j)}$, we have

$$\mathbb{E}[T^{(1)}] \leq \frac{2e^{-2\delta^2}}{1-e^{-2\delta^2}} + \sum_{j=1}^{\tilde{j}(\rho)}\sum_{k=1}^{K} \hat{\ell}_{\tilde{j}(\rho)+1,k}^{(j)} + 2K\tilde{j}(\rho)\log T + O(1)$$

$$\leq \sum_{j=1}^{\tilde{j}(\rho)}\sum_{k=1}^{K} \frac{27\log T}{2g_{\tilde{j}(\rho)+1}(\Delta_{\tilde{j}(\rho)+1,k}^{(j)})^2} + 2K\tilde{j}(\rho)\log T + O(1). \tag{36}$$

$\square$

Combining Lemma 4, Lemma 6, and Eq. (28), we have

$$\limsup_{T\to\infty} \frac{R_{\mathrm{UCB-ALP}}(T,B)}{\log T} \leq \Theta^{(a)} + \Theta_{\mathrm{nb}}^{(c)},$$

where

$$\Theta^{(a)} = \sum_{j=1}^{J}\sum_{k\neq k_j^*}\left(\frac{2}{\Delta_{j,k}^{(j)}} + 2\Delta_{j,k}^{(j)}\right),$$

$$\Theta_{\mathrm{nb}}^{(c)} = [\bar{u}^* + v(\rho)]\left\{\sum_{j=1}^{\tilde{j}(\rho)}\sum_{k=1}^{K}\frac{27}{2g_{\tilde{j}(\rho)+1}[\Delta_{\tilde{j}(\rho)+1,k}^{(j)}]^2} + \sum_{j=\tilde{j}(\rho)+2}^{J}\sum_{k=1}^{K}\frac{27}{2g_j[\Delta_{j,k}^{(\tilde{j}(\rho)+1)}]^2} + 2KJ\right\}.$$

This completes the proof of Part 1 in Theorem 2.

**Next, we discuss the bound of $R_{\mathrm{UCB-ALP}}^{(c)}(T,B)$ for the boundary cases**. The analysis is similar to the non-boundary cases with slight modification on the threshold.

We note that fundamentally, the context $\tilde{j}(\rho)+1$ for $\rho \neq q_j$ and the context $\tilde{j}(\rho)$ for $\rho = q_j$ are both the minimum context with positive probability in the static LP problem. Thus, we can define the context ranking events $\mathcal{E}_{\mathrm{rank},s}(t)$ $(0 \leq s \leq 2)$ similar to the analysis of Part 1, with $\tilde{j}(\rho)+1$ replaced by $\tilde{j}(\rho)$. Then, we have

$$R_{\mathrm{UCB-ALP}}^{(c)}(T,B) = \sum_{\tau=1}^{T}\mathbb{E}[\Delta v_\tau],$$

where

$$\mathbb{E}[\Delta v_\tau] = \sum_{s=0}^{2} \mathbb{E}[\Delta v_\tau, \mathcal{E}_{\text{rank},s}(T - \tau + 1)].$$

For the case of $s = 0$,

$$\mathbb{E}[\Delta v_\tau, \mathcal{E}_{\text{rank},0}(T - \tau + 1)] = \sum_{b=0}^{B} \mathbb{E}[\Delta v_\tau | b_\tau = b, \mathcal{E}_{\text{rank},0}(T - \tau + 1)]\mathbb{P}\{b_\tau = b, \mathcal{E}_{\text{rank},0}(T - \tau + 1)\}.$$

When $b/\tau \in [\rho - \delta, \rho + \delta]$ and $\mathcal{E}_{\text{rank},0}(T - \tau + 1)$ occurs, we have $\Delta v_\tau \leq (u_1^* - u_j^*)|\rho - b/\tau|$. Moreover, $\Delta v_\tau \leq v(\rho)$ under any condition. Thus,

$$\mathbb{E}[\Delta v_\tau, \mathcal{E}_{\text{rank},0}(T - \tau + 1)]$$
$$\leq u_1^* \mathbb{E}[|b_\tau/\tau - \rho|] + v(\rho) \sum_{b \notin [\rho - \delta, \rho + \delta]} \mathbb{P}\{b_\tau = b\}$$
$$\leq u_1^* \sqrt{\frac{\text{Var}(b_\tau)}{\tau^2}} + 2v(\rho)e^{-2\delta^2\tau}.$$

For the other cases of $s = 1, 2$, we have

$$\mathbb{E}[\Delta v_\tau, \mathcal{E}_{\text{rank},s}(T - \tau + 1)] \leq v(\rho)\mathbb{P}\{\mathcal{E}_{\text{rank},s}(T - \tau + 1)\}.$$

On the other hand, we extend Lemma 6 to the boundary cases:

$$\limsup_{T \to \infty} \frac{\mathbb{E}[T^{(1)}]}{\log T} \leq \sum_{j < \tilde{j}(\rho)} \sum_{k=1}^{K} \frac{27}{2g_{\tilde{j}(\rho)}[\Delta_{\tilde{j}(\rho),k}^{(j)}]^2} + 2K[\tilde{j}(\rho) - 1],$$

$$\limsup_{T \to \infty} \frac{\mathbb{E}[T^{(2)}]}{\log T} \leq \sum_{j \geq \tilde{j}(\rho)+1} \sum_{k=1}^{K} \frac{27}{2g_j[\Delta_{j,k}^{(\tilde{j}(\rho))}]^2} + 2K[J - \tilde{j}(\rho)],$$

where

$$g_j = \min\left\{\pi_j, \frac{1}{2}(\rho - q_{\tilde{j}(\rho)-1}), \frac{1}{2}(q_{\tilde{j}(\rho)+1} - \rho)\right\}.$$

Consequently, we can bound $R_{\text{UCB}-\text{ALP}}^{(c)}(T, B)$ by summing over the entire horizon and using the properties of $T^{(1)}$ and $T^{(2)}$. The conclusion of Part 2 of Theorem 2 then follows by adding the bound of $R_{\text{UCB}-\text{ALP}}^{(a)}(T, B)$ and $R_{\text{UCB}-\text{ALP}}^{(c)}(T, B)$.

## D  Two-Context Systems with Unit-Cost

As a special case, the oracle algorithm can be obtained for two-context systems with unit costs. When the context distribution and expected rewards are unknown, the oracle algorithm can be combined with the UCB method to achieve logarithmic regret under both boundary and non-boundary cases.

### D.1  Oracle Algorithm: Procrastinate-for-the-Better-context

When there are only two contexts, the oracle algorithm is trivial. Under the unit-cost assumption, skipping the worse context does not waste any opportunities if $b_\tau < \tau$. Thus, the agent can reserve budget for the better context, unless there is sufficient budget; i.e., we have the following algorithm:

**Procrastinate-for-the-Better (PB):** If $X_t = 1$ and $b_\tau > 0$, or if $b_\tau \geq \tau$, take action $A_t = k_{X_t}^*$; otherwise, $A_t = 0$.

We can verify that the above PB algorithm achieves the highest expected reward for any realization of the context arrival process. Thus, the PB algorithm is optimal in two-context systems. We note that the PB algorithm does not need to know the context distribution and only requires the ordering of the expected rewards. This property allows us to extend it to the case where the context distribution or expected rewards are unknown.

## D.2 UCB-PB: Logarithmic Regret Algorithm for Two-Context Bandits with Unit-Cost

When the context distribution and expected rewards are unknown, we propose the UCB-based Procrastinate-for-the-Better (UCB-PB) algorithm for solving the constrained contextual bandit problem with two contexts and unit costs.

---

**Algorithm 2** UCB-PB

---

**Input:** Time-horizon $T$, budget $B$;
**Init:** Remaining time $\tau = T$, remaining budget $b = B$;
    $C_{j,k}(0) = 0$, $\bar{u}_{j,k}(0) = 0$, $\hat{u}_{j,k}(0) = 1$, for all $j \in \mathcal{X}$
    and $k \in \mathcal{A}$; $\hat{u}_j^*(0) = 1$ for all $j \in \mathcal{X}$;
**for** $t = 1$ **to** $T$ **do**
    $k_j^*(t) \leftarrow \arg\max_k \hat{u}_{j,k}(t)$, $\forall j$;
    $\hat{u}_j^*(t) \leftarrow \hat{u}_{j,k_j^*(t)}^*(t)$, $\forall j$;
    $j^*(t) \leftarrow \arg\max_j \hat{u}_j^*(t)$;
    **if** $b \geq \tau$ **or** $(0 < b < \tau$ **and** $X_t = j^*(t))$ **then**
        Take action $k_{X_t}^*(t)$;
    **end if**
    Update $\tau$, $b$, $C_{j,k}(t)$, $\bar{u}_{j,k}(t)$, and $\hat{u}_{j,k}(t)$;
**end for**

---

As shown in Algorithm 2, the agent maintains UCB estimates $\hat{u}_{j,k}(t)$'s for the expected rewards of all context-action pairs. In each round, the agent implements the PB algorithm based on these estimates.

Next, we study the regret of the UCB-PB algorithm. We show that the UCB-PB algorithm achieves logarithmic regret for any given $\rho \in (0, 1)$.

**Theorem 3.** *For a constrained contextual bandit with unit-cost and two contexts, the UCB-PB algorithm achieves logarithmic regret as $T$ goes to infinity, i.e.,*

$$\limsup_{T \to \infty} \frac{R_{UCB\text{-}PB}(T, B)}{\log T} \leq \sum_{k=1}^{K} \left[ \frac{27}{2\pi_2 \Delta_{2,k}^{(1)}} + 2\Delta_{2,k}^{(1)} \right] + \sum_{j=1}^{2} \sum_{k \neq k_j^*} \left[ \frac{2}{\Delta_{j,k}^{(j)}} + 2\Delta_{j,k}^{(j)} \right].$$

*Proof.* The proof of Theorem 3 is similar to that of Theorem 2, while the analysis on the error events is even simpler. Note that the regret is defined as the difference between the expected total rewards achieved by the UCB-PB algorithm and the oracle algorithm. For the oracle algorithm, let $C_j^*(t) = \sum_{t'=1}^{t} \mathbb{1}\{X_{t'} = j, A_{t'} = k_j^*\}$ be the number of times that the context-action pair $(j, k_j^*)$ has been executed up to round $t$. For the UCB-PB algorithm, Recall that $C_{j,k}(t) = \sum_{t'=1}^{t} \mathbb{1}\{X_{t'} = j, A_{t'} = k_j\}$ is the number of times that the context-action pair $(j, k)$ has been executed up to round $t$, and let $C_j(t) = \sum_{k=1}^{K} C_{j,k}(t)$. Then the regret of UCB-PB can be expressed as

$$
\begin{aligned}
&R_{\text{UCB-PB}}(T, B) \\
=\ & \sum_{j=1}^{J} u_j^* \mathbb{E}[C_j^*(T)] - \sum_{j=1}^{J} \sum_{k=1}^{K} u_{j,k} \mathbb{E}[C_{j,k}(T)] \\
=\ & \sum_{j=1}^{J} \sum_{k=1}^{K} \Delta_{j,k}^{(j)} \mathbb{E}[C_{j,k}(T)] + \sum_{j=1}^{J} u_j^* \mathbb{E}\Big[C_j^*(T) - \sum_{k=1}^{K} C_{j,k}(T)\Big] \\
=\ & R_{\text{UCB-PB}}^{(a)}(T, B) + R_{\text{UCB-PB}}^{(c)}(T, B),
\end{aligned}
\tag{37}
$$

where $R_{\text{UCB-PB}}^{(a)}(T, B)$ is the regret due to action-ranking errors, i.e.,

$$R_{\text{UCB-PB}}^{(a)}(T, B) = \sum_{j=1}^{J} \sum_{k \neq k_j^*} \Delta_{j,k}^{(j)} \mathbb{E}[C_{j,k}(T)],$$

and $R_{\text{UCB-PB}}^{(c)}(T, B)$ is the regret due to context-ranking errors, i.e.,

$$R_{\text{UCB-PB}}^{(c)}(T, B) = \sum_{j=1}^{J} u_j^* \mathbb{E}\big[C_j^*(T) - \sum_{k=1}^{K} C_{j,k}(T)\big] = (u_1^* - u_2^*)\mathbb{E}\big[C_1^*(T) - C_1(T)\big]. \quad (38)$$

The expression of $R_{\text{UCB-PB}}^{(c)}(T, B)$ uses the fact that both the oracle algorithm and UCB-PB will exhaust their entire budget, i.e., $\sum_{j=1}^{J} C_j^*(T) = \sum_{j=1}^{J} C_j(T) = B$.

For $R_{\text{UCB-PB}}^{(a)}(T, B)$, we note that Lemma 4 also holds under UCB-PB, i.e.,

$$R_{\text{UCB-PB}}^{(a)}(T, B) \leq \sum_{j=1}^{J} \sum_{k \neq k_j^*} \left[ \left(\frac{2}{\Delta_{j,k}^{(j)}} + 2\Delta_{j,k}^{(j)}\right) \log T + 2\Delta_{j,k}^{(j)} \right]. \quad (39)$$

Next, we show that $R_{\text{UCB-PB}}^{(c)}(T, B)$ is also of order $O(\log T)$. Let $(\hat{X}_t, \hat{A}_t)$ be the context-action pair that has the highest UCB in round $t$. Moreover, let $\hat{C}_j(t)$ be the number of events that context $j$ has the maximum index up to round $t$, i.e., $\hat{C}_j(t) = \sum_{t'=1}^{t} \mathbb{1}(\hat{X}_t = j)$, and $\hat{C}_{j,k}(t)$ be the number of events that the context-action pair $(j,k)$ has the highest UCB up to round $t$, i.e., $\hat{C}_{j,k}(t) = \sum_{t'=1}^{t} \mathbb{1}(\hat{X}_t = j, \hat{A}_t = k)$. We show that the UCB-PB algorithm mistakes the suboptimal context as the optimal context for at most $O(\log T)$ times, i.e., $\mathbb{E}[\hat{C}_2(T)] = O(\log T)$, and then $\mathbb{E}\big[C_1^*(T) - C_1(T)\big] \leq \mathbb{E}[\hat{C}_2(T)] = O(\log T)$.

Specifically, consider the suboptimal context $j = 2$. For $1 \leq k \leq K$, we have

$$\mathbb{E}\big[\hat{C}_{2,k}(T)\big] \leq \hat{\ell}_{2,k}^{(1)} + \sum_{t=1}^{T} \mathbb{P}\{\hat{X}_t = 2, \hat{A}_t = k, b_\tau > 0, \hat{C}_{2,k}(t-1) \geq \hat{\ell}_{2,k}^{(1)}\},$$

where $\hat{\ell}_{2,k}^{(1)} = \frac{2 \log T}{\pi_2(1-\epsilon)\epsilon^2 (\Delta_{2,k}^{(1)})^2}$, and $\epsilon \in (0,1)$.

Based on Lemma 5, we have

$$\mathbb{P}\{C_{2,k}(t-1) < \pi_2(1-\epsilon)\hat{\ell}_{2,k}^{(1)}, b_\tau > 0, \hat{C}_{2,k}(t-1) \geq \hat{\ell}_{2,k}^{(1)}\} \leq e^{-2\epsilon^2 \ell_{2,k}} \leq T^{-4}. \quad (40)$$

Thus,

$$\mathbb{P}\{\hat{X}_t = 2, \hat{A}_t = k, \hat{C}_{2,k}(t-1) \geq \hat{\ell}_{2,k}^{(1)}, b_\tau > 0\}$$
$$\leq \mathbb{P}\{\hat{X}_t = 2, \hat{A}_t = k, C_{2,k}(t-1) \geq \pi_2(1-\epsilon)\hat{\ell}_{2,k}^{(1)}\}$$
$$\quad + \mathbb{P}\{C_{2,k}(t-1) < \pi_2(1-\epsilon)\hat{\ell}_{2,k}^{(1)}, \hat{C}_{2,k}(t-1) \geq \hat{\ell}_{2,k}^{(1)}, b_\tau > 0\}$$
$$\leq \mathbb{P}\{\hat{u}_{2,k}^*(t) > \hat{u}_{1,k_1^*}(t) | C_{2,k}(t-1) \geq \pi_2(1-\epsilon)\hat{\ell}_{2,k}^{(1)}\}$$
$$\quad + \mathbb{P}\{C_{2,k}(t-1) < \pi_2(1-\epsilon)\hat{\ell}_{2,k}^{(1)}, \hat{C}_{2,k}(t-1) \geq \hat{\ell}_{2,k}^{(1)}, b_\tau > 0\}$$
$$\leq 2t^{-1} + T^{-4},$$

where the last inequality results from Lemma 3 (note that for $j = 2$, $\hat{u}_{2,k}(t) < \hat{u}_{1,k_1^*}(t) \leq \hat{u}_1^*(t)$) and Eq. (40).

Summing over all actions, we have

$$\mathbb{E}[\hat{C}_2(T)] = \sum_{k=1}^{K} \mathbb{E}\big[\hat{C}_{2,k}(T)\big] \leq \sum_{k=1}^{K} \hat{\ell}_{2,k}^{(1)} + \sum_{t=1}^{T}\sum_{k=1}^{K}(2t^{-1} + T^{-4})$$

$$= \sum_{k=1}^{K} \left[ \frac{2}{\pi_2(1-\epsilon)\epsilon^2 (\Delta_{2,k}^{(1)})^2} + 2 \right] \log T + O(1).$$

Consequently,

$$
\begin{aligned}
\mathbb{E}[C_1^*(T) - C_1(T)] &\leq \sum_{k=1}^{K} \left[ \frac{2}{\pi_2(1-\epsilon)\epsilon^2(\Delta_{2,k}^{(1)})^2} + 2 \right] \log T + O(1) \\
&= \sum_{k=1}^{K} \left[ \frac{27}{2\pi_2(\Delta_{2,k}^{(1)})^2} + 2 \right] \log T + O(1). \quad (41)
\end{aligned}
$$

The last equality is obtained by letting $\epsilon = 2/3$. Combining Eqs. (39), (38), and (41), and using the fact that $u_1^* - u_2^* \leq u_1^* - u_{2,k}$ for all $k$, we can obtain the conclusion of Theorem 3. $\qquad\square$

## E   Constrained Contextual Bandits with Unknown Context Distribution

In this section, we relax the assumption of known context distribution and study unit-cost systems with unknown context distribution. Since the arrival of contexts is independent of the actions taken by the agent. a natural idea is to implement the ALP or UCB-ALP algorithm based on the empirical distribution as follows:

**EALP and UCB-EALP Algorithms:** the agent maintains the empirical distribution of the contexts, denoted by $\hat{\boldsymbol{\pi}}_t = (\hat{\pi}_{1,t}, \hat{\pi}_{2,t}, \ldots, \hat{\pi}_{J,t})$, where $\hat{\pi}_{j,t} = \frac{1}{t}\sum_{t'=1}^{t} \mathbb{1}(X_{t'} = j)$. In each round, the agent executes the ALP (when the expected rewards are known) or UCB-ALP (when the expected rewards are unknown) algorithms with the context distribution $\boldsymbol{\pi}$ in $\mathcal{LP}_{\tau,b}$ replaced by $\hat{\boldsymbol{\pi}}_t$. These algorithms are referred to as Empirical ALP (EALP) and UCB-EALP, respectively.

As we can see from the numerical simulations in Appendix G, the above EALP and UCB-EALP algorithms have similar performance as ALP and UCB-ALP, respectively. However, the regret analysis for these algorithms is challenging because the empirical distribution introduces complex temporal dependency since the empirical distribution depends on the context arrivals in all the past rounds, which makes it difficult to analyze the evolution of the remaining budget. Thus, we focus on the non-boundary cases and consider truncated version of EALP and UCB-EALP. Specifically, we study algorithms that stop updating the empirical distribution from the $T_1$-th (will be defined later) round and use the fixed estimate $\hat{\boldsymbol{\pi}}_{T_1}$ for the remaining rounds, which are referred to as EALP2 (as shown in Algorithm 3) and UCB-EALP2, respectively. We focus on the EALP2 algorithm for the case where the expected rewards are known, while the properties of UCB-EALP2 can be obtained by similar techniques in the analysis of UCB-ALP combined with the properties of EALP2.

---

**Algorithm 3** EALP2

> **Input:** Time horizon $T$, budget $B$, learning stage length $T_1$, and expected rewards $u_j^*$'s;
> **Init:** $\tau = T$; $b = B$; $\hat{\pi}_{j,0} = 0, \forall j$;
> **for** $t = 1$ **to** $T$ **do**
>     **if** $t \leq T_1$ **then**
>         $\hat{\pi}_{j,t} \leftarrow \frac{(t-1)\hat{\pi}_{j,t-1} + \mathbb{1}(X_t=j)}{t}, \forall j$;
>     **end if**
>     **if** $b > 0$ **then**
>         Obtain the probabilities $p_j(b/\tau)$'s by solving $\mathcal{LP}_{\tau,b}$ with $\boldsymbol{\pi}$ replaced by $\hat{\boldsymbol{\pi}}_t$.
>         Take action $k_{X_t}^*$ with probability $p_{X_t}(b/\tau)$.
>     **end if**
> **end for**

---

Now we show that for a sufficiently large $T$ and an appropriate chosen $T_1$, the EALP2 algorithm achieves similar performance as ALP in the non-boundary cases. Let $\delta = \min\{q_{\tilde{j}(\rho)+1} - \rho, \rho - q_{\tilde{j}(\rho)}\}$ be the gap between $\rho$ and the boundaries. The following theorem shows that EALP2 achieves $O(1)$ regret with appropriately chosen $T$ and $T_1$.

**Theorem 4.** *Given a fixed $\rho \in (0,1)$, $\rho \neq q_j$, $j = 1, 2, \ldots, J-1$. If $T_1 = 16J^2 \log^3 T/\delta^2$ and $T$ satisfies $\log^3 T/T \leq \delta^3/(64J^2)$, then the regret of EALP2 satisfies $R_{\text{EALP2}}(T, B) = O(1)$.*

Note that here we assume $\delta$ is known for the simplicity of presentation. When considering practical scenarios where $\delta$ is unknown, we can obtain a lower confidence bound of $\delta$ as follows. At round $t$, let $\hat{q}_{j,t} = \sum_{j'=1}^{j} \hat{\pi}_{j,t}$ be the empirical estimate of the cumulative probability. Further, let $\tilde{j}_t'(\rho) =$

$\max\{j : \hat{q}_{j,t} \leq \rho\}$ be the threshold under the empirical estimate. Let $\hat{\delta}_t = \min\{\hat{q}_{\tilde{j}'_t(\rho)+1} - \rho, \rho - \hat{q}_{\tilde{j}'_t(\rho)}\}$ and $\check{\delta}_t = \frac{1}{2}\hat{\delta}_t$. Then $\check{\delta}_t$ is a lower confidence bound of $\delta$ with $\mathbb{P}\{\delta \geq \check{\delta}_t\} \geq 1 - e^{-2\check{\delta}_t^2 t}$. We choose $T_1$ which is the smallest $t$ such that $e^{-2\check{\delta}_t^2 t} \leq \frac{1}{T^2}$ and $t \geq 16J^2 \log^3 T/\check{\delta}_t^2$. Then the following analysis holds, while the regret due to the event that $\delta \geq \check{\delta}_t$ will be $O(1)$. Moreover, such a $t$ will appear with high probability after $64J^2 \log^3 T/\delta^2$ rounds for the non-boundary cases, and not appear with high probability for the boundary cases.

Similar to the non-boundary cases in Theorem 1, the key idea of proving Theorem 4 is to show that under EALP2, the average remaining budget $b_\tau/\tau$ will not cross the boundaries with high probability. To achieve this, we examine the expectation of $b_\tau/\tau$ and its concentration properties under EALP2.

**Step 1: Estimation error of $\hat{\boldsymbol{\pi}}_{T_1}$.** Let $\alpha = \delta/(4J \log T)$. According to Hoeffding-Chernoff bound, we have

$$\mathbb{P}\{|\hat{\pi}_{j,T_1} - \pi_j| \leq \alpha, \forall j\} \geq 1 - \frac{2J}{T^2}. \tag{42}$$

**Step 2: Bound on the expectation of $b_\tau/\tau$.**

**Lemma 7.** *Assume $|\hat{\pi}_{j,T_1} - \pi_j| \leq \alpha$ for all $j$. Then, for all $T_1 \leq t \leq T$, the expectation of the average remaining budget satisfies*

$$|\mathbb{E}[b_\tau/\tau] - \rho| \leq \frac{\delta}{2}, \tag{43}$$

*where $\tau = T - t + 1$.*

*Proof.* First, we note that the average remaining budget $b_\tau/\tau$ is close to the initial value $\rho$ at round $T_1$, because we can verify that for all $t \leq T_1$,

$$\rho - \frac{\delta}{4} \leq \frac{B - T_1}{T - T_1 + 1} \leq \frac{b_\tau}{\tau} \leq \frac{B}{T - T_1 + 1} \leq \rho + \frac{\delta}{4}. \tag{44}$$

Now we show by mathematical induction that, if $|\hat{\pi}_{j,T_1} - \pi_j| \leq \alpha$ for all $j$, then $|\mathbb{E}[b_\tau/\tau] - \rho| \leq J\alpha \sum_{\tau'=T-T_1+1}^{\tau} \frac{1}{\tau'} + \frac{\delta}{4}$ for $\tau \leq T - T_1 + 1$ (i.e., $t \geq T_1$).

Specifically, for $t = T_1$, we have $\left|\mathbb{E}[\frac{b_{T-T_1+1}}{T-T_1+1}] - \rho\right| \leq \frac{\delta}{4}$ according to Eq. (44). For any given $t \geq T_1$, we have $\tau = T - t + 1$, and

$$\mathbb{E}[b_{\tau-1}|b_\tau = b]$$
$$= b - \left(\frac{b/\tau - \sum_{j \leq \tilde{j}(b/\tau)} \hat{\pi}_{j,T_1}}{\hat{\pi}_{\tilde{j}(b/\tau)+1,T_1}} \pi_{\tilde{j}(b/\tau)+1} + \sum_{j \leq \tilde{j}(b/\tau)} \pi_j\right)$$
$$= b - b/\tau + \frac{b/\tau - \sum_{j \leq \tilde{j}(b/\tau)} \hat{\pi}_{j,T_1}}{\hat{\pi}_{\tilde{j}(b/\tau)+1,T_1}}(\hat{\pi}_{\tilde{j}(b/\tau)+1,T_1} - \pi_{\tilde{j}(b/\tau)+1}) + \sum_{j \leq \tilde{j}(b/\tau)} (\hat{\pi}_{j,T_1} - \pi_j).$$

Note that $0 < \frac{b/\tau - \sum_{j \leq \tilde{j}(b/\tau)} \hat{\pi}_{j,T_1}}{\hat{\pi}_{\tilde{j}(b/\tau)+1,T_1}} < 1$. Thus, $\left|\mathbb{E}[b_{\tau-1}|b_\tau = b] - \frac{b(\tau-1)}{\tau}\right| \leq J\alpha$, implying that $\left|\mathbb{E}[\frac{b_{\tau-1}}{\tau-1}|b_\tau = b] - \frac{b}{\tau}\right| \leq \frac{J\alpha}{\tau-1}$. If $\left|\mathbb{E}[b_\tau/\tau] - \rho\right| \leq J\alpha \sum_{\tau'=T-T_1+1}^{\tau} \frac{1}{\tau'} + \frac{\delta}{4}$ for $2 \leq \tau \leq T - T_1 + 1$, then $\left|\mathbb{E}[b_{\tau-1}/(\tau-1)] - \rho\right| \leq J\alpha \sum_{\tau'=T-T_1+1}^{\tau} \frac{1}{\tau'} + \frac{J\alpha}{\tau-1} + \frac{\delta}{4}$.

The conclusion of Lemma 7 then follows because $|\mathbb{E}[b_\tau/\tau] - \rho| \leq J\alpha \sum_{\tau'=T-T_1+1}^{\tau} \frac{1}{\tau'} + \frac{\delta}{4} \leq J\alpha \log T + \frac{\delta}{4} \leq \frac{\delta}{2}$. $\qquad\square$

**Step 3: Concentration of $b_\tau/\tau$.** The next lemma shows the concentration of the average remaining budget $b_\tau/\tau$.

**Lemma 8.** *Assume $|\hat{\pi}_{j,T_1} - \pi_j| \leq \alpha$ for all $j$. The average remaining budget $b_\tau/\tau$ in round $t = T - \tau + 1$ satisfies*

$$\mathbb{P}\{|\frac{b_\tau}{\tau} - \mathbb{E}[\frac{b_\tau}{\tau}]| \geq \delta/4\} \leq 2\exp\left(-\frac{\delta^2\tau}{32}\right). \tag{45}$$

To show the concentration of $b_\tau / \tau$, we first use the coupling argument to show the following lemma and then use the method of averaged bounded differences [23].

**Lemma 9.** *Assume $|\hat{\pi}_{j,T_1} - \pi_j| \leq \alpha$ for all $j$. The remaining budget $b_\tau$ in round $t = T - \tau + 1$ satisfies*

$$\left| \mathbb{E}[b_\tau | \mathbf{Z}_{t'-1}, Z_{t'} = 1] - \mathbb{E}[b_\tau | \mathbf{Z}_{t'-1}, Z_{t'} = 0] \right| \leq 2 \left( \frac{\tau}{T - t'} \right)^{1-\sigma}, \quad T_1 \leq t' < t. \tag{46}$$

*where $\mathbf{Z}_{t'-1} = (Z_1, Z_2, \ldots, Z_{t'-1})$ and $\sigma = 1 - \min_j \frac{\pi_j}{\pi_j + \alpha}$.*

*Proof.* We bound the difference by constructing a coupling $\mathcal{M}$ of the two conditional distributions $(\cdot | \mathbf{Z}_{t'-1}, Z_{t'} = 1)$ and $(\cdot | \mathbf{Z}_{t'-1}, Z_{t'} = 0)$. Let $\zeta_{t'+1}, \zeta_{t'+2}, \ldots, \zeta_{T-\tau}$ and $\zeta'_{t'+1}, \zeta'_{t'+2}, \ldots, \zeta'_{T-\tau}$ be the pair of random variables in the coupling $\mathcal{M}$. We construct the coupling as follows:

**Coupling:** We generate the value of $\zeta_{t''}$'s and $\zeta'_{t''}$'s sequentially. For each $t'' > t'$, let $\tilde{b}_{T-t''+1} = B - 1 - \sum_{s=1}^{t'-1} Z_s - \sum_{s=t'+1}^{t''-1} \zeta_s$ and $\tilde{b}'_{T-t''+1} = B - \sum_{s=1}^{t'-1} Z_s - \sum_{s=t'+1}^{t''-1} \zeta'_s$ be the remaining budgets in round $t''$ corresponding to the pair of random variables. For $\zeta_{t''}$, We pick its value randomly with distribution $\mathbb{P}\{\zeta_{t''} = 1\} = h\left( \frac{\tilde{b}_{T-t''+1}}{T-t''+1} \right)$ and $\mathbb{P}\{\zeta_{t''} = 0\} = 1 - h\left( \frac{\tilde{b}_{T-t''+1}}{T-t''+1} \right)$, where $h(\rho)$ is the probability that one unit of budget will be consumed under EALP2 when the average remaining budget is $\rho$, i.e.,

$$h(\rho) = \frac{\rho - \sum_{j' \leq \tilde{j}(\rho)} \hat{\pi}_{j,T_1}}{\hat{\pi}_{\tilde{j}(\rho)+1,T_1}} \pi_{\tilde{j}(\rho)+1} + \sum_{j' \leq \tilde{j}(\rho)} \pi_j. \tag{47}$$

For $\zeta'_{t''}$, we generate its value conditioned on $\zeta_{t''}$. If $\tilde{b}'_{T-t''+1} = \tilde{b}_{T-t''+1}$, then $\zeta'_{t''} = \zeta_{t''}$. If $\tilde{b}'_{T-t''+1} = \tilde{b}_{T-t''+1} + 1$, then

$$\mathbb{P}\{\zeta'_{t''} = \zeta_{t''} | \tilde{b}'_{T-t''+1} = \tilde{b}_{T-t''+1} + 1\} = 1,$$

$$\mathbb{P}\{\zeta'_{t''} = 1 | \tilde{b}'_{T-t''+1} = \tilde{b}_{T-t''+1} + 1, \zeta_{t''} = 1\} = 1,$$

$$\mathbb{P}\{\zeta'_{t''} = 1 | \tilde{b}'_{T-t''+1} = \tilde{b}_{T-t''+1} + 1, \zeta_{t''} = 0\} = \frac{h\left( \frac{\tilde{b}'_{T-t''+1}}{T-t''+1} \right) - h\left( \frac{\tilde{b}_{T-t''+1}}{T-t''+1} \right)}{1 - h\left( \frac{\tilde{b}_{T-t''+1}}{T-t''+1} \right)}.$$

Note that according to the above construction, $\tilde{b}'_{T-t''+1} - \tilde{b}_{T-t''+1}$ could only be 0 or 1. We can verify that the marginals satisfy

$$(\zeta_{t''}, t'' > t') \sim (Z_{t''}, t'' > t' | \mathbf{Z}_{t'-1}, Z_{t'} = 1),$$

and

$$(\zeta'_{t''}, t'' > t') \sim (Z_{t''}, t'' > t' | \mathbf{Z}_{t'-1}, Z_{t'} = 0).$$

From the construction of the coupling, we know that $\tilde{b}'_\tau - \tilde{b}_\tau = 1$ if and only if $\zeta_{t''} = \zeta'_{t''}$ for all $t' < t'' \leq T - \tau$. Thus,

$$\left| \mathbb{E}[b_\tau | \mathbf{Z}_{t'-1}, Z_{t'} = 1] - \mathbb{E}[b_\tau | \mathbf{Z}_{t'-1}, Z_{t'} = 0] \right|$$
$$= \mathbb{P}\{\zeta_{t''} = \zeta'_{t''}, t' < t'' \leq T - \tau\}$$
$$= \prod_{t''=t'+1}^{T-\tau} \mathbb{P}\{\zeta_{t''} = \zeta'_{t''} | \zeta_s = \zeta'_s, t' < s \leq t'' - 1\}. \tag{48}$$

We show that each term in Eq. (48) can be bounded as follows.

**Lemma 10.** *The coupling $\mathcal{M}$ satisfies*

$$\mathbb{P}\{\zeta_{t''} = \zeta'_{t''} | \zeta_s = \zeta'_s, t' < s \leq t'' - 1\} \leq 1 - \frac{1 - \sigma}{T - t'' + 1},$$

*where $\sigma = 1 - \min_j \frac{\pi_j}{\pi_j + \alpha}$.*

*Proof.* Conditioned on $\zeta_s = \zeta'_s, t' < s \le t'' - 1$, we have $\tilde{b}'_{T-t''+1} = \tilde{b}_{T-t''+1} + 1$, and $\zeta_{t''} \ne \zeta'_{t''}$ i.f.f. $\zeta'_{t''} = 0$ and $\zeta_{t''} = 1$. Thus,

$$
\begin{aligned}
\mathbb{P}\{\zeta_{t''} = \zeta'_{t''} | \zeta_s = \zeta'_s, t' < s \le t'' - 1\} &= 1 - \frac{h\big(\frac{\tilde{b}'_{T-t''+1}}{T-t''+1}\big) - h\big(\frac{\tilde{b}_{T-t''+1}}{T-t''+1}\big)}{1 - h\big(\frac{\tilde{b}_{T-t''+1}}{T-t''+1}\big)} \Big[1 - h\big(\frac{\tilde{b}_{T-t''+1}}{T-t''+1}\big)\Big] \\
&= 1 - \Big[h\big(\frac{\tilde{b}'_{T-t''+1}}{T-t''+1}\big) - h\big(\frac{\tilde{b}_{T-t''+1}}{T-t''+1}\big)\Big].
\end{aligned}
\tag{49}
$$

To prove Lemma 10, it suffices to show that for any $b$ and $\tau$ satisfying $b \le \tau - 1$, we have $h\big(\frac{b+1}{\tau}\big) - h\big(\frac{b}{\tau}\big) \ge \gamma/\tau$, where $\gamma = \min_j \frac{\pi_j}{\pi_j + \alpha}$.

Specifically, from the definition of $h(\cdot)$ in Eq. (47), we know that if $\tilde{j}\big(\frac{b+1}{\tau}\big) = \tilde{j}\big(\frac{b}{\tau}\big)$, we have $h\big(\frac{b+1}{\tau}\big) - h\big(\frac{b}{\tau}\big) = \frac{\pi_{\tilde{j}(\frac{b}{\tau})}}{\hat{\pi}_{\tilde{j}(\frac{b}{\tau}),T_1}} \cdot \frac{1}{\tau} \ge \gamma/\tau$.

If $\tilde{j}\big(\frac{b+1}{\tau}\big) > \tilde{j}\big(\frac{b}{\tau}\big)$, we have $\frac{b}{\tau} - \sum_{j \le \tilde{j}(\frac{b}{\tau})} \hat{\pi}_{j,T_1} < \hat{\pi}_{\tilde{j}(\frac{b}{\tau})+1,T_1}$ and $\frac{b+1}{\tau} - \sum_{j \le \tilde{j}(\frac{b+1}{\tau})} \hat{\pi}_{j,T_1} > 0$. Therefore,

$$
\begin{aligned}
&h\big(\frac{b+1}{\tau}\big) - h\big(\frac{b}{\tau}\big) \\
&= \sum_{j=\tilde{j}(\frac{b}{\tau})+1}^{\tilde{j}(\frac{b+1}{\tau})} \pi_j + \frac{\frac{b+1}{\tau} - \sum_{j \le \tilde{j}(\frac{b+1}{\tau})} \hat{\pi}_{j,T_1}}{\hat{\pi}_{\tilde{j}(\frac{b+1}{\tau})+1,T_1}} \pi_{\tilde{j}(\frac{b+1}{\tau})+1} - \frac{\frac{b}{\tau} - \sum_{j \le \tilde{j}(\frac{b}{\tau})} \hat{\pi}_{j,T_1}}{\hat{\pi}_{\tilde{j}(\frac{b}{\tau})+1,T_1}} \pi_{\tilde{j}(\frac{b}{\tau})+1} \\
&= \sum_{j=\tilde{j}(\frac{b}{\tau})+2}^{\tilde{j}(\frac{b+1}{\tau})} \pi_j + \frac{\pi_{\tilde{j}(\frac{b}{\tau})+1}}{\hat{\pi}_{\tilde{j}(\frac{b}{\tau})+1,T_1}} \Big[\hat{\pi}_{\tilde{j}(\frac{b}{\tau})+1,T_1} - \big(\frac{b}{\tau} - \sum_{j \le \tilde{j}(\frac{b}{\tau})} \hat{\pi}_{j,T_1}\big)\Big] + \frac{\pi_{\tilde{j}(\frac{b+1}{\tau})+1}}{\hat{\pi}_{\tilde{j}(\frac{b+1}{\tau})+1,T_1}} \Big[\frac{b+1}{\tau} - \sum_{j \le \tilde{j}(\frac{b+1}{\tau})} \hat{\pi}_{j,T_1}\Big] \\
&\ge \sum_{j=\tilde{j}(\frac{b}{\tau})+2}^{\tilde{j}(\frac{b+1}{\tau})} \frac{\pi_j}{\hat{\pi}_{j,T_1}} \hat{\pi}_{j,T_1} + \gamma\Big(\frac{1}{\tau} - \sum_{j=\tilde{j}(\frac{b}{\tau})+2}^{\tilde{j}(\frac{b+1}{\tau})} \hat{\pi}_{j,T_1}\Big) \ge \gamma/\tau.
\end{aligned}
\tag{50}
$$

$\square$

Using Lemma 10, we have

$$
\begin{aligned}
\big|\mathbb{E}[b_\tau | \mathbf{Z}_{t'-1}, Z_{t'} = 1] - \mathbb{E}[b_\tau | \mathbf{Z}_{t'-1}, Z_{t'} = 0]\big| &\le \prod_{t''=t'+1}^{T-\tau} \big(1 - \frac{1-\sigma}{T-t''+1}\big) \\
&\overset{(a)}{=} \frac{\tau+\sigma}{T-t'} \prod_{s=1}^{T-t'-\tau-1} \big(1 + \frac{\sigma}{\tau+s}\big) \\
&\overset{(b)}{\le} \frac{\tau+\sigma}{T-t'} \big(\frac{T-t'-1}{\tau}\big)^\sigma \\
&\le 2\big(\frac{\tau}{T-t'}\big)^{1-\sigma}.
\end{aligned}
$$

Equality (a) is obtained by merging the numerator of each term with the denominator of the next term. Inequality (b) is true because $\sigma < 1$, and

$$
\log \prod_{s=1}^{T-t'-\tau-1} \big(1 + \frac{\sigma}{\tau+s}\big) = \sum_{s=1}^{T-t'-\tau-1} \log\big(1 + \frac{\sigma}{\tau+s}\big) \le \sum_{s=1}^{T-t'-\tau-1} \frac{\sigma}{\tau+s} \le \sigma \log\big(\frac{T-t'-1}{\tau}\big).
$$

$\square$

To use the method of averaged bounded differences [23], we note that

$$
\sum_{t'=T_1}^{T-\tau} \Big[2\big(\frac{\tau}{T-t'}\big)^{1-\sigma}\Big]^2 \le 4\tau^{2-2\alpha} \cdot \Big[\frac{1}{\tau^{1-2\sigma}} - \frac{1}{(T-T_1+1)^{1-2\sigma}}\Big] \le 4\tau.
\tag{51}
$$

Then, according to Corollary 5.1 in [23] and Lemma 9, we have

$$\mathbb{P}\big\{|b_\tau - \mathbb{E}[b_\tau]| \geq \delta\tau/4\big\} \leq 2\exp\big(-\frac{2(\delta\tau/4)^2}{4\tau}\big) = 2\exp\big(-\frac{\delta^2\tau}{32}\big).$$

implying Eq. (45) in Lemma 8.

**Step 4: Upper bound of $R_{\mathrm{EALP2}}(T,B)$.** Now we bound the regret $R_{\mathrm{EALP2}}(T,B)$ using the results obtained in the previous steps. We analyze the event of "boundary-crossing" in round $t$, denoted as $\mathcal{E}_{\mathrm{cross},t}$, which is the event that $\tilde{j}(b_\tau/\tau) \neq \tilde{j}(\rho)$. The event of "boundary-crossing" may happen when the estimates of empirical distribution is inaccurate or the average remaining budget $b_\tau/\tau$ deviates far from $\rho$. We study the probability of $\mathcal{E}_{\mathrm{cross},t}$ for $t \leq T_1$ and $t > T_1$, respectively.

For $t \leq T_1$, the average remaining budget satisfies $\rho - \delta/4 \leq b_\tau/\tau \leq \rho + \delta/4$, as discussed in Step 2. The event $\mathcal{E}_{\mathrm{cross},t}$ may occur only when there is some $j$ such that $|\hat{\pi}_{j,t} - \pi_j| \geq \delta/(4J)$. Thus,

$$\mathbb{P}\{\mathcal{E}_{\mathrm{cross},t}\} \leq \mathbb{P}\{\exists j, |\hat{\pi}_{j,t} - \pi_j| \geq \delta/(4J)\} \leq 2J\exp(-\delta^2 t/8J^2), \quad t \leq T_1. \tag{52}$$

For $t > T_1$, if the empirical distribution $|\hat{\pi}_{j,T_1} - \pi_j| \leq \alpha$ $(< \delta/(4J)$ for sufficiently large $T)$ for all $j$, then the average remaining budget satisfies $\mathbb{P}\{|\frac{b_\tau}{\tau} - \mathbb{E}[\frac{b_\tau}{\tau}]| \geq 3\delta/4\} \leq 2\exp\big(-\frac{\delta^2\tau}{32}\big)$ due to Lemma 7 and Lemma 8. Thus,

$$
\begin{aligned}
\mathbb{P}\{\mathcal{E}_{\mathrm{cross},t}\} &\leq \mathbb{P}\{\exists j, |\hat{\pi}_{j,T_1} - \pi_j| \geq \alpha\} + \mathbb{P}\{|\frac{b_\tau}{\tau} - \mathbb{E}[\frac{b_\tau}{\tau}]| \geq 3\delta/4 | \forall j, |\hat{\pi}_{j,T_1} - \pi_j| \geq \alpha\} \\
&\leq \frac{2J}{T^2} + 2\exp\big(-\frac{\delta^2\tau}{32}\big), \quad t > T_1.
\end{aligned} \tag{53}
$$

Now we bound the expectation of $C_j(T)$, i.e., the number of executions under context $j$.

For $j \leq \tilde{j}(\rho)$,

$$
\begin{aligned}
\mathbb{E}[C_j(T)] &= \mathbb{E}\Big[\sum_{t=1}^{T} \mathbb{1}(X_t = j, A_t = k_j^*)\Big] \\
&\geq \sum_{t=1}^{T} \mathbb{P}\{X_t = j, A_t = k_j^* | \neg\mathcal{E}_{\mathrm{cross},t}\}\mathbb{P}\{\neg\mathcal{E}_{\mathrm{cross},t}\} \\
&= \sum_{t=1}^{T} \pi_j\big(1 - \mathbb{P}\{\mathcal{E}_{\mathrm{cross},t}\}\big) \\
&\geq \pi_j T - \sum_{t=1}^{T_1} 2J\exp(-\delta^2 t/8J^2) - \sum_{t=T_1+1}^{T}\Big[\frac{2J}{T^2} + 2\exp\big(-\frac{\delta^2\tau}{32}\big)\Big] = \pi_j T + O(1),
\end{aligned}
$$

and

$$\mathbb{E}[C_j(T)] \leq \pi_j T,$$

Similarly, for $j > \tilde{j}(\rho) + 1$, we have

$$\mathbb{E}[C_j(T)] \leq \sum_{t=1}^{T} \mathbb{P}\{X_t = j, A_t = k_j^* | \mathcal{E}_{\mathrm{cross},t}\}\mathbb{P}\{\mathcal{E}_{\mathrm{cross},t}\} = O(1). \tag{54}$$

For $j = \tilde{j}(\rho) + 1$, we have

$$\mathbb{E}[C_j(T)] = \mathbb{E}[B - b_0] - \sum_{j \neq \tilde{j}(\rho)+1} \mathbb{E}[C_j(t)] \geq B - T\sum_{j \leq \tilde{j}(\rho)} \pi_j - O(1) = \big(\rho - \sum_{j \leq \tilde{j}(\rho)} \pi_j\big)T - O(1).$$

We complete the proof of Theorem 4 by summing over all contexts:

$$U_{\mathrm{EALP2}}(T,B) = \sum_{j=1}^{J} \mu_j^* \mathbb{E}[C_j(T)] \geq T\tilde{v}(\rho) - O(1) = \widehat{U}(T,B) - O(1).$$

# F  Constrained Contextual Bandits with Heterogeneous Costs

In this section, we consider the case where the cost for each action $k$ under context $j$ is fixed at $c_{j,k}$, which may be different for different $j$ and $k$. We discuss how to use the insight from unit-cost systems in heterogeneous-cost systems.

## F.1  Approximation of the Oracle Algorithm

Similar to unit-cost systems, we first study the case with known statistics. We generalize the upper bound and the ALP algorithm in Section 3 to general-cost systems.

### F.1.1  Upper Bound

With known statistics, the agent knows the context distribution $\pi_j$'s, the costs $c_{j,k}$'s, and the expected rewards $u_{j,k}$'s. In heterogeneous-cost systems, the quality of a context-action pair $(j, k)$ is roughly captured by the normalized reward, denoted by $\eta_{j,k} = u_{j,k}/c_{j,k}$. However, unlike the unit-cost case, the agent *cannot* only focus on the "best" action with highest normalized reward, i.e., $k_j^* = \arg\max_k \eta_{j,k}$, when making a decision under context $j$. This is because there may exist another action $k$ such that $\eta_{j,k} < \eta_{j,k_j^*}$, but $u_{j,k} > u_{j,k_j^*}$ (and of course, $c_{j,k} > c_{j,k_j^*}$). If there is sufficient budget allocated for context $j$, then the agent may take action $k$ to maximize the expected reward. Therefore, the agent needs to consider all actions under each context. Let $p_{j,k}$ be the probability that action $k$ is taken under context $j$. We define the following LP problem:

$$(\mathcal{LP}'_{T,B}) \quad \text{maximize} \quad \sum_{j=1}^{J} \pi_j \sum_{k=1}^{K} p_{j,k} u_{j,k}, \tag{55}$$

$$\text{subject to} \quad \sum_{j=1}^{J} \pi_j \sum_{k=1}^{K} p_{j,k} c_{j,k} \leq B/T, \tag{56}$$

$$\sum_{k=1}^{K} p_{j,k} \leq 1, \; \forall j, \tag{57}$$

$$p_{j,k} \in [0, 1].$$

The above LP problem $\mathcal{LP}'_{T,B}$ can be solved efficiently by optimization tools. Let $\hat{v}(\rho)$ be the maximum value of $\mathcal{LP}'_{T,B}$. Similar to Lemma 1, we can show that $T\hat{v}(\rho)$ is an upper bound of the expected total reward, i.e., $T\hat{v}(\rho) \geq U^*(T, B)$.

To obtain insight from the solution of $\mathcal{LP}'_{T,B}$, we derive an explicit representation for the solution by analyzing the structure of $\mathcal{LP}'_{T,B}$. Note that there are two types of (non-trivial) constraints in $\mathcal{LP}'_{T,B}$, one is the "inter-context" budget constraint (56), the other is the "intra-context" constraint (57). These constraints can be decoupled by first allocating budget for each context, and then solving a subproblem with the allocated budget constraint for each context. Specifically, let $\rho_j$ be the budget allocated to context $j$, then $\mathcal{LP}'_{T,B}$ can be decomposed as follows:

$$\text{maximize} \quad \sum_{j=1}^{J} \pi_j \hat{v}_j(\rho_j),$$

$$\text{subject to} \quad \sum_{j=1}^{J} \pi_j \rho_j \leq B/T,$$

where

$$(\mathcal{SP}_j) \quad v_j(\rho_j) = \text{maximize} \quad \sum_{k=1}^{K} p_{j,k} u_{j,k}, \tag{58}$$

$$\text{subject to} \quad \sum_{k=1}^{K} p_{j,k} c_{j,k} \leq \rho_j, \tag{59}$$

$$\sum_{k=1}^{K} p_{j,k} \leq 1, \tag{60}$$

$$p_{j,k} \in [0,1].$$

Next, by analyzing sub-problem $\mathcal{SP}_j$, we show that some actions can be deleted without affecting the performance, i.e., the probability is 0 in the optimal solution.

**Lemma 11.** *For any given $\rho_j \geq 0$, there exists an optimal solution of $\mathcal{SP}_j$, i.e., $\boldsymbol{p}_j^* = (p_{j,1}^*, p_{j,2}^*, \ldots, p_{j,K}^*)$, satisfies:*
*(1) For $k_1$, if there exists another action $k_2$, such that $\eta_{j,k_1} \leq \eta_{j,k_2}$ and $u_{k_1} \leq u_{k_2}$, then $p_{j,k_1}^* = 0$;*
*(2) For $k_1$, if there exists two actions $k_2$ and $k_3$, such that $\eta_{j,k_2} \leq \eta_{j,k_1} \leq \eta_{j,k_3}$, $u_{j,k_2} \geq u_{j,k_1} \geq u_{j,k_3}$, and $\frac{u_{j,k_1} - u_{j,k_3}}{c_{j,k_1} - c_{j,k_3}} \leq \frac{u_{j,k_2} - u_{j,k_3}}{c_{j,k_2} - c_{j,k_3}}$, then $p_{j,k_1}^* = 0$.*

Intuitively, the first part of Lemma 11 shows that if an action has small normalized and original expected reward, then it can be removed. The second part of Lemma 11 shows that if an action has small normalized expected reward and medium original expected reward, but the increasing rate is smaller than another action with larger expected reward, then it can also be removed.

*Proof.* The key idea of this proof is that, if the conditions is satisfied, and there is a feasible solution $\boldsymbol{p}_j = (p_{j,1}, p_{j,2}, \ldots, p_{j,K})$ such that $p_{j,k_1} > 0$, then we can construct another feasible solution $\boldsymbol{p}_j'$ such that $p_{j,k_1}' = 0$, without reducing the objective value $v_j(\rho_j)$.

We first prove part (1). Under the conditions of part (1), if $\boldsymbol{p}_j$ is a feasible solution of $\mathcal{SP}_j$ with $p_{j,k_1} > 0$, then consider another solution $\boldsymbol{p}_j'$, where $p_{j,k}' = p_{j,k}$ for $k \notin \{k_1, k_2\}$, $p_{j,k_1}' = 0$, and $p_{j,k_2}' = p_{j,k_2} + p_{j,k_1} \min\{\frac{c_{j,k_1}}{c_{j,k_2}}, 1\}$. Then, we can verify that $\boldsymbol{p}_j'$ is a feasible solution of $(\mathcal{SP}_j)$, and the objective value under $\boldsymbol{p}_j'$ is no less than that under $\boldsymbol{p}_j$.

For the second part, if the conditions are satisfied and $p_{j,k_1} > 0$, then we construct a new solution $\boldsymbol{p}_j'$ by re-allocating the budget consumed by action $k_1$ to actions $k_2$ and $k_3$, without violating the constraints. Specifically, we set the probability the same as the original solution for other actions, i.e., $p_{j,k}' = p_{j,k}$ for $k \notin \{k_1, k_2, k_3\}$, and set $p_{j,k_1}' = 0$ for action $k_1$. For $k_2$ and $k_3$, to maximize the objective function, we would like to allocate as much budget as possible to $k_3$ unless there is remaining budget. Therefore, we set $p_{j,k_2}' = p_{j,k_2}$ and $p_{j,k_3}' = p_{j,k_3} + \frac{p_{j,k_1} c_{j,k_1}}{c_{j,k_3}}$, if $\sum_{k \neq k_1} p_{j,k} + \frac{p_{j,k_1} c_{j,k_1}}{c_{j,k_3}} \leq 1$; or, $p_{j,k_2}' = p_{j,k_2} + \frac{p_{j,k_1} c_{j,k_1} - (1 - \sum_{k \neq k_1} p_{j,k}) c_{j,k_3}}{c_{j,k_2} - c_{j,k_3}}$ and $p_{j,k_3}' = p_{j,k_3} + \frac{(1 - \sum_{k \neq k_1} p_{j,k}) c_{j,k_2} - p_{j,k_1} c_{j,k_1}}{c_{j,k_2} - c_{j,k_3}}$, if $\sum_{k \neq k_1} p_{j,k} + \frac{p_{j,k_1} c_{j,k_1}}{c_{j,k_3}} > 1$. We can verify that $\boldsymbol{p}_j$ satisfies the constraints of $(\mathcal{SP}_j)$ but the objective value is no less than that under $\boldsymbol{p}_j$. $\qquad \square$

With Lemma 11, the agent can ignore some actions that will obviously be allocated with zero probability under a given context $j$. We call the set of the remaining actions as *candidate set* for context $j$, denoted as $\mathcal{A}_j$. We propose an algorithm to construct the candidate action set for context $j$, as shown in Algorithm 4.

For context $j$, assume that the candidate set $\mathcal{A}_j = \{k_{j,1}, k_{j,2}, \ldots, k_{j,K_j}\}$ has been sorted in descending order of their normalized rewards, i.e., $\eta_{j,k_{j,1}} \geq \eta_{j,k_{j,2}} \geq \ldots \geq \eta_{j,k_{j,K_j}}$. From Algorithm 4, we know that $u_{j,k_{j,1}} < u_{j,k_{j,2}} < \ldots < u_{j,k_{j,K_j}}$, and $c_{j,k_{j,1}} < c_{j,k_{j,2}} < \ldots < c_{j,k_{j,K_j}}$.

The agent now only needs to consider the actions in the candidate set $\mathcal{A}_j$. To decouple the "intra-context" constraint (57), we introduce the following transformation:

$$p_{j,k_{j,a}} = \begin{cases} \tilde{p}_{j,k_{j,a}} - \tilde{p}_{j,k_{j,a+1}}, & \text{if } 1 \leq a \leq K_j - 1, \\ \tilde{p}_{j,k_{j,K_j}}, & \text{if } a = K_j, \end{cases}$$

---

**Algorithm 4** Find Candidate Set for Context $j$

---

**Input:** $c_{j,k}$'s, $u_{j,k}$'s, for all $1 \leq k \leq K$;
**Output:** $\mathcal{A}_j$;
**Init:** $\mathcal{A}_j = \{1, 2, \ldots, K\}$ ;
Calculate normalized rewards: $\eta_{j,k} = u_{j,k}/c_{j,k}$;
Sort actions in descending order of their normalized rewards:

$$\eta_{j,k_1} \geq \eta_{j,k_2} \geq \ldots \geq \eta_{j,k_K}.$$

**for** $a = 2$ **to** $K$ **do**
  **if** $\exists a' < a$ such that $u_{j,k_a} \leq u_{j,k_{a'}}$ **then**
    $\mathcal{A}_j = \mathcal{A}_j \backslash \{k_a\}$;
  **end if**
**end for**
$a = 1$;
**while** $a \leq K - 1$ **do**
  Find the action with highest increasing rate:

$$a^* = \operatorname*{arg\,max}_{a':a'>a, k_{a'} \in \mathcal{A}_j} \frac{u_{j,k_{a'}} - u_{j,k_a}}{c_{j,k_{a'}} - c_{j,k_a}}.$$

  Remove the actions in between:

$$\mathcal{A}_j = \mathcal{A}_j \backslash \{k_{a'} : a < a' < a^*\}.$$

  Move to the next candidate action: $a = a^*$;
**end while**

---

where $\tilde{p}_{j,k_{j,a}} \in [0,1]$, and $\tilde{p}_{j,k_{j,a}} \geq \tilde{p}_{j,k_{j,a+1}}$ for $1 \leq a \leq K_j - 1$. Substituting the transformations into $(\mathcal{SP}_j)$ and reorganize it as

$$
\begin{aligned}
(\widetilde{\mathcal{SP}_j}) \ \ \text{maximize} \ & \sum_{a=1}^{K_j} \tilde{p}_{j,k_{j,a}} \tilde{u}_{j,k_{j,a}}, \\
\text{subject to} \ & \sum_{a=1}^{K_j} \tilde{p}_{j,k_{j,a}} \tilde{c}_{j,k_{j,a}} \leq \rho_j, \\
& \tilde{p}_{j,k_{j,a}} \geq \tilde{p}_{j,k_{j,a+1}}, \ 1 \leq a \leq K_j - 1, \qquad (61) \\
& \tilde{p}_{j,k_{j,a}} \in [0,1], \ \forall a,
\end{aligned}
$$

where

$$
\tilde{u}_{j,k_{j,a}} = \begin{cases} u_{j,k_{j,1}}, & \text{if } a = 1, \\ u_{j,k_{j,a}} - u_{j,k_{j,a-1}}, & \text{if } 2 \leq a \leq K_j, \end{cases}
$$

$$
\tilde{c}_{j,k_{j,a}} = \begin{cases} c_{j,k_{j,1}}, & \text{if } a = 1, \\ c_{j,k_{j,a}} - c_{j,k_{j,a-1}}, & \text{if } 2 \leq a \leq K_j. \end{cases}
$$

Next, we show that the constraint (61) can indeed be removed. For each $k_{j,a}$, we can view $\tilde{c}_{j,k_{j,a}}$ and $\tilde{u}_{j,k_{j,a}}$ as the cost and expected reward of a virtual action. Let $\tilde{\eta}_{j,k_{j,a}} = \tilde{u}_{j,k_{j,a}}/\tilde{c}_{j,k_{j,a}}$ be the normalized expected reward of virtual action $k_{j,a}$. For $a = 1$, using $\frac{u_{j,k_{j,1}}}{c_{j,k_{j,1}}} \geq \frac{u_{j,k_{j,2}}}{c_{j,k_{j,2}}}$, we can show that $\tilde{\eta}_{j,k_{j,1}} \geq \tilde{\eta}_{j,k_{j,2}}$. For $2 \leq a \leq K_j - 1$, using $\frac{u_{j,k_{j,a}} - u_{j,k_{j,a-1}}}{c_{j,k_{j,a}} - c_{j,k_{j,a-1}}} \geq \frac{u_{j,k_{j,a+1}} - u_{j,k_{j,a-1}}}{c_{j,k_{j,a+1}} - c_{j,k_{j,a-1}}}$, we can show that $\tilde{\eta}_{j,k_{j,a}} \geq \tilde{\eta}_{j,k_{j,a+1}}$. In other words, we can verify that $\tilde{\eta}_{j,k_{j,1}} \geq \tilde{\eta}_{j,k_{j,2}} \geq \ldots \geq \tilde{\eta}_{j,k_{j,K_j}}$. Thus, without constraint (61), the optimal solution $\tilde{\boldsymbol{p}}_j^* = [\tilde{p}_{j,k_1}^*, \tilde{p}_{j,k_2}^*, \ldots, \tilde{p}_{j,k_{K_j}}^*]$ automatically satisfies $\tilde{p}_{j,k_1}^* \geq \tilde{p}_{j,k_2}^* \geq \ldots \geq \tilde{p}_{j,k_{K_j}}^*$. Hence, we can remove the constraint (61), and thus decouple the probability constraint under a context.

With the above transformations, we can thus rewrite the global LP problem

$$(\widetilde{\mathcal{LP}}'_{T,B}) \text{ maximize } \sum_{j=1}^{J}\sum_{a=1}^{K_j} \pi_j \tilde{p}_{j,k_{j,a}} \tilde{u}_{j,k_{j,a}},$$

$$\text{subject to } \sum_{j=1}^{J}\sum_{a=1}^{K_j} \pi_j \tilde{p}_{j,k_{j,a}} \tilde{c}_{j,k_{j,a}} \le B/T,$$

$$\tilde{p}_{j,k_{j,a}} \in [0,1], \ \forall j, \text{ and } 1 \le a \le K_j.$$

The solution of $\widetilde{\mathcal{LP}}'_{T,B}$ follows a threshold structure. We sort all context-(virtual-)action pairs $(j, k_a)$ in descending order of their normalized expected reward. Let $j^{(i)}, k^{(i)}$ be the context index and action index of the $i$-th pair, respectively. Namely, $\tilde{\eta}_{j^{(1)},k^{(1)}} \ge \tilde{\eta}_{j^{(2)},k^{(2)}} \ge \ldots \ge \tilde{\eta}_{j^{(M)},k^{(M)}}$, where $M = \sum_{j=1}^{J} K_j$ is the total number of candidate actions for all contexts. Define a threshold corresponding to $\rho = B/T$,

$$\tilde{i}(\rho) = \max\{i : \sum_{i'=1}^{i} \pi_{j^{(i')}} \tilde{c}_{j^{(i')},k^{(i')}} \le \rho\}, \tag{62}$$

where $\rho = B/T$ is the average budget. We can verify that the following solution is optimal for $\widetilde{\mathcal{LP}}'_{T,B}$:

$$\tilde{p}_{j^{(i)},k^{(i)}}(\rho) = \begin{cases} 1, & \text{if } 1 \le i \le \tilde{i}(\rho), \\ \frac{\rho - \sum_{i'=1}^{\tilde{i}(\rho)} \pi_{j^{(i')}} \tilde{c}_{j^{(i')},k^{(i')}}}{\pi_{j^{(\tilde{i}(\rho)+1)}} \tilde{c}_{j^{(\tilde{i}(\rho)+1)},k^{(\tilde{i}(\rho)+1)}}}, & \text{if } i = \tilde{i}(\rho) + 1, \\ 0, & \text{if } i > \tilde{i}(\rho) + 1. \end{cases}$$

Then, the optimal solution of $\widetilde{\mathcal{LP}}'_{T,B}$ can be calculated using the reverse transformation from $\tilde{p}_{j,k}(\rho)$'s to $p_{j,k}(\rho)$'s

### F.1.2 ALP Algorithm

Similar to unit-cost systems, the ALP algorithm replaces the average constraint $B/T$ in $\mathcal{LP}'_{T,B}$ with the average remaining budget $b_\tau/\tau$, and obtains probability $p_{j,k}(b_\tau/\tau)$. Under context $j$, the ALP algorithm take action $k$ with probability $p_{j,k}(b_\tau/\tau)$.

Unlike unit-cost systems, the remaining budget $b_\tau$ does not follow any classic distribution in heterogeneous-cost systems. However, we can show that the concentration property still holds for this general case by using the method of averaged bounded differences [23].

**Lemma 12.** *For $0 < \delta < 1$, there exists a positive number $\kappa$, such that under the ALP algorithm, the remaining budget $b_\tau$ satisfies*

$$\mathbb{P}\{b_\tau > (\rho + \delta)\tau\} \le e^{-\kappa\delta^2\tau},$$
$$\mathbb{P}\{b_\tau < (\rho - \delta)\tau\} \le e^{-\kappa\delta^2\tau}.$$

*Proof.* We prove the lemma using the method of averaged bounded differences [23]. The process is similar to Section 7.1 in [23], except that we consider the remaining budget and the successive differences of the remaining budget are bounded by $c_{\max}$.

Specifically, let $\tilde{c}_{t'}$, $1 \le t' \le T$ be the budget consumed under ALP, and let $\tilde{\mathbf{c}}_{t'} = (\tilde{c}_1, \tilde{c}_2, \ldots, \tilde{c}_{t'})$. Then the remaining budget at round $t$ (the remaining time $\tau = T - t + 1$), i.e., $b_{T-t+1}$ is a function of $\tilde{\mathbf{c}}_t$. We note that under ALP, the expectation of the ratio between the remaining budget and the remaining time does not change, i.e., for any $b \le \sum_{j=1} \pi_j c_j^*$ (here $c_j^* = \max_k c_{j,k}$), if $b_\tau = b$, then $\mathbb{E}[b_{\tau-1}/(\tau-1)] = b/\tau$. Thus, we can verify that for any $1 \le t' \le t$, we have

$$\mathbb{E}[b_{T-t+1}|\tilde{\mathbf{c}}_{t'}] = b_{T-t'+1} - \frac{b_{T-t'+1}}{T-t'+1}(t-t'). \tag{63}$$

Note that $\Delta b = b_{T-t'+2} - b_{T-t'+1} \leq c_{\max}$ and $b_{T-t'+2} \geq -c_{\max}$, we have

$$\left| \mathbb{E}[b_{T-t+1}|\tilde{c}_{t'}] - \mathbb{E}[b_{T-t+1}|\tilde{c}_{t'-1}] \right|$$
$$\leq \max_{0 \leq \Delta b \leq c_{\max}} \left\{ \left| \Delta b - \frac{b_{T-t'+2}}{T-t'+2} \right| \right\} \frac{T-t+1}{T-t'+1}$$
$$\leq \frac{2c_{\max}(T-t+1)}{T-t'+1}. \tag{64}$$

Moreover,

$$\sum_{t'=1}^{t} \left[ \frac{2c_{\max}(T-t+1)}{T-t'+1} \right]^2$$
$$= 4c_{\max}^2(T-t+1)^2 \sum_{t'=1}^{t} \frac{1}{(T-t'+1)^2}$$
$$= 4c_{\max}^2(T-t+1)^2 \sum_{\tau'=T-t+1}^{T} \frac{1}{(\tau')^2}$$
$$\approx 4c_{\max}^2(T-t+1)^2 \int_{T-t+1}^{T} \frac{1}{(\tau')^2} d\tau'$$
$$= 4c_{\max}^2(T-t+1)\frac{t-1}{T}. \tag{65}$$

According to Theorem 5.3 in [23], and noting $\tau = T - t + 1$, $\mathbb{E}[b_\tau] = \rho\tau$, we have

$$\mathbb{P}\{b_\tau > \mathbb{E}[b_\tau] + \delta\tau\} \leq e^{-\frac{2T(\delta\rho\tau)^2}{4c_{\max}^2(T-t+1)(t-1)}} \leq e^{-\frac{T\delta^2 B^2\tau}{2c_{\max}^2 T^2(t-1)}} \leq e^{-\frac{\delta^2\rho^2}{2c_{\max}^2}\tau}, \tag{66}$$

and similarly,

$$\mathbb{P}\{b_\tau < \mathbb{E}[b_\tau] - \delta\tau\} \leq e^{-\frac{\delta^2\rho^2}{2c_{\max}^2}\tau}, \tag{67}$$

Choosing $\kappa = \frac{\rho^2}{2c_{\max}^2}$ concludes the proof. $\square$

Then, using similar methods in Section 3, we can show that the generalized ALP algorithm achieves $O(1)$ regret in non-boundary cases, and $O(\sqrt{T})$ regret in boundary cases, where the boundaries are now defined as $Q_i = \sum_{i'=1}^{i} \pi_{j^{(i')}} \tilde{c}_{j^{(i')},k^{(i')}}$.

### F.2 $\epsilon$-First ALP Algorithm

When the expected rewards are unknown, it is difficult to combine UCB method with the proposed ALP for general systems. As a special case, when all actions have the same cost under a given context, i.e., $c_{j,k} = c_j$ for all $k$ and $j$, the normalized expected reward $\eta_{j,k}$ represents the quality of action $k$ under context $j$. In this case, the candidate set for each context only contains one action, which is the action with the highest expected reward. Thus, the ALP algorithm for the known statistics case is simple. When the expected rewards are unknown, we can extend the UCB-ALP algorithm by managing the UCB for the normalized expected rewards.

When the costs for different actions under the same context are heterogeneous, it is difficult to combine ALP with the UCB method since the ALP algorithm in this case not only requires the ordering of $\eta_{j,k}$'s, but also the ordering of $u_{j,k}$'s and the ratios $\frac{u_{j,k_1} - u_{j,k_2}}{c_{j,k_1} - c_{j,k_2}}$. We propose an $\epsilon$-First ALP Algorithm that explores and exploits separately: the agent takes actions under all contexts in the first $\epsilon(T)$ rounds to estimate the expected rewards, and runs ALP based on the estimates in the remaining $T - \epsilon(T)$ rounds.

For the ease of exposition, we assume $c_{j,k_1} \neq c_{j,k_2}$ for any $j$ and $k_1 \neq k_2$ [2], and let $\Delta_{\min}^{(c)}$ be the minimal difference, i.e.,

$$\Delta_{\min}^{(c)} = \min_{\substack{j \in \mathcal{X} \\ k_1,k_2 \in \{0\} \cup \mathcal{A}}} \{|c_{j,k_1} - c_{j,k_2}|\}.$$

**Algorithm 5** $\epsilon$-First ALP

---

**Input:** Time horizon $T$, budget $B$, exploration stage length $\epsilon(T)$, and $c_{j,k}$'s, for all $j$ and $k$;
**Init:** Remaining budget $b = B$;
$C_{j,k} = 0$, $\bar{u}_{j,k} = 0$;
**for** $t = 1$ **to** $\epsilon(T)$ **do**
    **if** $b > 0$ **then**
        Take action $A_t = \arg\min_{k \in \mathcal{A}} C_{X_t,k}$ (with random tie-breaking);
        Observe the reward $Y_{A_t,t}$;
        Update counter $C_{X_t,A_t} = C_{X_t,A_t} + 1$; update remaining budget $b = b - c_{X_t,A_t}$;
        Update the reward estimate:

$$\bar{u}_{X_t,A_t} = \frac{(C_{X_t,A_t} - 1)\bar{u}_{X_t,A_t} + Y_{A_t,t}}{C_{X_t,A_t}}.$$

    **end if**
**end for**
**for** $t = \epsilon(T) + 1$ **to** $T$ **do**
    Remaining time $\tau = T - t + 1$;
    **if** $b > 0$ **then**
        Obtain the probabilities $p_{j,k}(b/\tau)$'s by solving the problem $(\mathcal{LP}'_{\tau,b})$ with $u_{j,k}$ replaced by $\bar{u}_{j,k}$;
        Take action $k$ with probability $p_{X_t,k}(b/\tau)$;
        Remaining budget $b = b - c_{X_t,A_t}$;
    **end if**
**end for**

---

Let $\xi_{j,k_1,k_2} = \frac{u_{j,k_1} - u_{j,k_2}}{c_{j,k_1} - c_{j,k_2}}$ for $j \in \mathcal{X}$, $k_1, k_2 \in \{0\} \cup \mathcal{A}$, and $k_1 \neq k_2$ (recall that $u_{j,0} = 0$ and $c_{j,0} = 0$ for the dummy action), $\bar{\xi}_{j,k_1,k_2}$ be its estimate at the end of the exploration stage, i.e., $\bar{\xi}_{j,k_1,k_2} = \frac{\bar{u}_{j,k_1} - \bar{u}_{j,k_2}}{c_{j,k_1} - c_{j,k_2}}$. Let $\Delta_{\min}^{(\xi)}$ be the minimal difference between any $\xi_{j_1,k_{11},k_{12}}$ and $\xi_{j_2,k_{21},k_{22}}$, i.e.,

$$\Delta_{\min}^{(\xi)} = \min_{\substack{j_1,j_2 \in \mathcal{X} \\ k_{11},k_{12},k_{21},k_{22} \in \{0\} \cup \mathcal{A}}} \{|\xi_{j_1,k_{11},k_{12}} - \xi_{j_2,k_{21},k_{22}}|\}.$$

Moreover, let $\pi_{\min} = \min_{j \in \mathcal{X}} \pi_j$ and let $\Delta^* = \Delta_{\min}^{(c)} \Delta_{\min}^{(\xi)}$. Then, the following lemma states that under $\epsilon$-First ALP with a sufficiently large $\epsilon(T)$, the agent will obtain a correct ordering of $\xi_{j,k_1,k_2}$'s with high probability at the end of the exploration stage.

**Lemma 13.** *Let $0 < \delta < 1$. Under $\epsilon$-First ALP, if*

$$\epsilon(T) = \left\lceil \frac{K}{(1-\delta)\pi_{\min}} + \log T \max\left\{ \frac{1}{\delta^2}, \frac{16K}{(1-\delta)\pi_{\min}(\Delta^*)^2} \right\} \right\rceil,$$

*then for any contexts $j_1, j_2 \in \mathcal{X}$, and actions $k_{11}, k_{12}, k_{21}, k_{22} \in \{0\} \cup \mathcal{A}$, if $\xi_{j_1,k_{11},k_{12}} < \xi_{j_2,k_{21},k_{22}}$, then at the end of the $\epsilon(T)$-th round, we have*

$$\mathbb{P}\left\{ \bar{\xi}_{j_1,k_{11},k_{12}} \geq \bar{\xi}_{j_2,k_{21},k_{22}} \right\} \leq (J+4)T^{-2}.$$

*Moreover, the agent ranks all the $\xi_{j,k_1,k_2}$'s correctly with probability no less than $1-(4K+1)JT^{-2}$.*

*Proof.* We first analyze the number of executions for each context-action pair $(j, k)$ in the exploration stage. Let $N_j = \sum_{t=1}^{\epsilon(T)} \mathbb{1}(X_t = j)$ be the number of occurrences of context $j$ up to round $\epsilon(T)$. Recall that the contexts $X_t$ arrive i.i.d. in each round. Thus, using Hoeffding-Chernoff Bound

for each context $j$, we have

$$\mathbb{P}\left\{\forall j \in \mathcal{X}, N_j \geq (1-\delta)\pi_j \epsilon(T)\right\}$$

$$\geq 1 - \sum_{j=1}^{J} \mathbb{P}\left\{N_j < (1-\delta)\pi_j \epsilon(T)\right\}$$

$$\geq 1 - Je^{-2\delta^2 \epsilon(T)}$$

$$\geq 1 - Je^{-2\log T}$$

$$= 1 - JT^{-2} \tag{68}$$

On the other hand, the lower bound $(1-\delta)\pi_j\epsilon(T) \geq K + \frac{16K\log T}{(\Delta^*)^2}$. From the implementation of the exploration stage in Algorithm 5, we know that if $N_j \geq (1-\delta)\pi_j\epsilon(T)$, then

$$C_{j,k} \geq \lfloor 1 + \frac{16\log T}{(\Delta^*)^2} \rfloor \geq \frac{16\log T}{(\Delta^*)^2}, \quad \forall k \in \mathcal{A}. \tag{69}$$

Therefore,

$$\mathbb{P}\left\{\forall j \in \mathcal{X}, \forall k \in \mathcal{A}, C_{j,k} \geq \frac{16\log T}{(\Delta^*)^2}\right\}$$

$$\geq 1 - JT^{-2} \tag{70}$$

Next, we study the relationship between the estimates $\bar{\xi}_{j_1,k_{11},k_{12}}$ and $\bar{\xi}_{j_2,k_{21},k_{22}}$ at the end of the exploration stage. We note that

$$\bar{\xi}_{j_1,k_{11},k_{12}} \geq \bar{\xi}_{j_2,k_{21},k_{22}}$$

$$\Leftrightarrow \left(\bar{\xi}_{j_1,k_{11},k_{12}} - \xi_{j_1,k_{11},k_{12}} - \frac{\xi_{j_2,k_{21},k_{22}} - \xi_{j_1,k_{11},k_{12}}}{2}\right)$$

$$- \left(\bar{\xi}_{j_2,k_{21},k_{22}} - \xi_{j_2,k_{21},k_{22}} + \frac{\xi_{j_2,k_{21},k_{22}} - \xi_{j_1,k_{11},k_{12}}}{2}\right) \geq 0$$

$$\Leftrightarrow \left(\frac{\bar{u}_{j_1,k_{11}} - u_{j_1,k_{11}}}{c_{j_1,k_{11}} - c_{j_1,k_{12}}} - \frac{\xi_{j_2,k_{21},k_{22}} - \xi_{j_1,k_{11},k_{12}}}{4}\right)$$

$$- \left(\frac{\bar{u}_{j_1,k_{12}} - u_{j_1,k_{12}}}{c_{j_1,k_{11}} - c_{j_1,k_{12}}} + \frac{\xi_{j_2,k_{21},k_{22}} - \xi_{j_1,k_{11},k_{12}}}{4}\right)$$

$$- \left(\frac{\bar{u}_{j_2,k_{21}} - u_{j_2,k_{21}}}{c_{j_2,k_{21}} - c_{j_2,k_{22}}} + \frac{\xi_{j_2,k_{21},k_{22}} - \xi_{j_1,k_{11},k_{12}}}{4}\right)$$

$$+ \left(\frac{\bar{u}_{j_2,k_{22}} - u_{j_2,k_{22}}}{c_{j_2,k_{21}} - c_{j_1,k_{22}}} - \frac{\xi_{j_2,k_{21},k_{22}} - \xi_{j_1,k_{11},k_{12}}}{4}\right) \geq 0. \tag{71}$$

Thus, for the event $\bar{\xi}_{j_1,k_{11},k_{12}} \geq \bar{\xi}_{j_2,k_{21},k_{22}}$ to be true, we require that at least one term (with the sign) in the last inequation above is no less than zero. Conditioned on $C_{j,k} \geq \frac{16\log T}{(\Delta^*)^2}$, we can bound the probability of each term according to the Hoeffding-Chernoff bound. For example, for the first term, we have

$$\mathbb{P}\{\frac{\bar{u}_{j_1,k_{11}} - u_{j_1,k_{11}}}{c_{j_1,k_{11}} - c_{j_1,k_{12}}} - \frac{\xi_{j_2,k_{21},k_{22}} - \xi_{j_1,k_{11},k_{12}}}{4} \geq 0$$

$$|C_{j_1,k_{11}} \geq \frac{16\log T}{(\Delta^*)^2}\}$$

$$\leq \mathbb{P}\{\bar{u}_{j_1,k_{11}} \geq u_{j_1,k_{11}} + \frac{\Delta^*}{4}|C_{j_1,k_{11}} \geq \frac{16\log T}{(\Delta^*)^2}\}$$

$$\leq e^{-2\log T} = T^{-2}.$$

The conclusion then follows by considering the event $\{C_{j,k} \geq \frac{16\log T}{(\Delta^*)^2}, \forall j \in \mathcal{X}, \forall k \in \mathcal{X}\}$ and its negation. $\qquad\square$

**Theorem 5.** *Let* $0 < \delta < 1$. *Under $\epsilon$-First ALP, if*

$$\epsilon(T) \geq \frac{K}{(1-\delta)\pi_{\min}} + \log T \max\left\{\frac{1}{\delta^2}, \frac{16K}{(1-\delta)\pi_{\min}(\Delta^*)^2}\right\},$$

*then the regret of $\epsilon$-First ALP satisfies:*
*1) if $\rho = B/T \neq Q_i$, then $R_{\epsilon-\mathrm{FirstALP}}(T, B) = O(\log T)$;*

*2) if $\rho = B/T = Q_i$, then $R_{\epsilon-\mathrm{FirstALP}}(T, B) = O(\sqrt{T})$.*

*Proof.* (Sketch) The key idea of proving this theorem is considering the event where the $\xi_{j,k_1,k_2}$'s are ranked correctly and its negation. When the $\xi_{j,k_1,k_2}$'s are ranked correctly, we can use the properties of the ALP algorithm with modification on the time horizon and budget (subtracting the time and budget in the exploration stage, which is $O(\log T)$); otherwise, if the agent obtains a wrong ranking results, the regret is bounded as $O(1)$ because the probability is $O(T^{-2})$ and the reward in each round is bounded.

### F.3 Deciding $\epsilon(T)$ without Prior Information

In Theorem 5, the agent requires the value of $\Delta^*$ (in fact $\Delta_{\min}^{(\xi)}$ because $\Delta_{\min}^{(c)}$ is known) to calculate $\epsilon(T)$. This is usually impractical since the expected rewards are unknown *a priori*. Thus, without the knowledge of $\Delta_{\min}^{(\xi)}$, we propose a Confidence Level Test (CLT) algorithm for deciding when to end the exploration stage.

Specifically, assume $\Delta_{\min}^{(\xi)} > 0$ and is unknown by the agent. In each round of the exploration stage, the agent tries to solve the problem $(\mathcal{LP}'_{\tau,b})$ with $u_{j,k}$ replaced by $\bar{u}_{j,k}$ using comparison, i.e., using Algorithm 4 and sorting the virtual actions. For each comparison, the agent tests the confidence level according to Algorithm 6. If all comparisons pass the test, i.e., `flagSucc = true` for all comparisons, then the agent ends the exploration stage and starts the exploitation stage.

---

**Algorithm 6** Confidence Level Test (CLT)

---

**Input:** Time horizon $T$, estimates $\bar{\xi}_{j_1,k_{11},k_{12}}$, $\bar{\xi}_{j_2,k_{21},k_{22}}$, number of executions $C_{j_1,k_{11}}$, $C_{j_1,k_{12}}$, $C_{j_2,k_{21}}$, and $C_{j_2,k_{22}}$;
**Output:** `flagSucc`;
**Init:** `flagSucc = false`,
$\quad\quad \Delta' = \frac{\Delta_{\min}^{(c)}(\bar{\xi}_{j_1,k_{11},k_{12}} - \bar{\xi}_{j_2,k_{21},k_{22}})}{2}$;
**if** $e^{-2(\Delta')^2 \min\{C_{j_1,k_{11}}, C_{j_1,k_{12}}\}} \leq T^{-2}$ & $e^{-2(\Delta')^2 \min\{C_{j_2,k_{21}}, C_{j_2,k_{22}}\}} \leq T^{-2}$ **then**
$\quad$ `flagSucc = true`;
**end if**
**return** `flagSucc`;

---

Next, we show that the $\epsilon$-First policy with CLT will achieve $O(\log T)$ regret except for the boundary cases, where it achieves $O(\sqrt{T})$ regret. On one hand, according to Hoeffding-Chernoff bound, if all comparisons pass the confidence level test, then with probability at least $1 - JK^2T^{-2}$, the algorithm obtains the correct rank and provide a right solution for the problem $(\mathcal{LP}'_{\tau,b})$. On the other hand, because $\Delta^* > 0$, from the analysis in the previous section, we know that the exploration stage will end within $O(\log T)$ rounds with high probability. Therefore, the expected regret is the same as that in the case with known $\Delta_{\min}^{(\xi)}$.

## G  Numerical Experiments

In this section, we evaluate the regret of the proposed algorithms through numerical simulations. We study the performance of the proposed algorithms here for unit-cost systems as the parameter setting is relatively simple to control while providing us useful insights. The performance in heterogeneous-cost systems is similar as we have shown theoretically, and omitted here. In the case with known statistics, we compare the proposed PB (two-context case) and ALP algorithms with Fixed LP (FLP) algorithm that uses a fixed average budget constraint $B/T$ since both [17] and [20] use fixed average budget constraint. Then, the UCB-based FLP, i.e., UCB-FLP, is evaluated in the case without

knowledge of expected rewards. We also evaluate algorithms for the case without knowledge of context distribution. When the context distribution is unknown to the agent, we use the Empirical ALP (EALP) algorithm, that uses the empirical distribution (histogram) of context for making decisions, in the case with known expected rewards. Then, the UCB-based EALP is proposed for the case without knowledge of expected rewards. The results are averaged from 5,000 independent runs of the simulations.

## G.1 Two-Context Systems

(a)  (b)  (c)

Figure 1: Comparison of algorithms for the two-context systems with perfect knowledge ($\pi_1 = 0.4, \pi_2 = 0.6$), (a) $\rho = 0.39$, (b) $\rho = 0.4$, (c) $\rho = 0.41$.

We first consider a two-context scenario with $K = 3$ arms and Bernoulli rewards: the context distribution vector is $\boldsymbol{\pi} = [0.4, 0.6]$, the expected rewards are $\boldsymbol{u}_1 = 0.8 \times [1/3, 2/3, 1]$ for context 1, and $\boldsymbol{u}_2 = 0.4 \times [1/3, 2/3, 1]$ for context 2. The boundary is $q_1 = \pi_1 = 0.4$ and we study the cases with normalized budget $\rho = 0.39, 0.4$, and $0.41$, respectively.

Figure 1 shows the regret of different algorithms in the case with known expected rewards. In the non-boundary cases (i.e., $\rho = 0.39, 0.41$), the ALP algorithm achieves near optimal performance. Even without the knowledge of context distribution, the EALP algorithm performs much better than FLP. In the boundary case, i.e., $\rho = 0.4$, the regret of ALP increases with $T$ but is still lower than that of FLP. The EALP algorithm achieves higher regret than ALP and FLP due to the empirical distribution errors.

(a)  (b)  (c)

Figure 2: Comparison of algorithms for the two-context systems without perfect knowledge ($\pi_1 = 0.4, \pi_2 = 0.6$), (a) $\rho = 0.39$, (b) $\rho = 0.4$, (c) $\rho = 0.41$.

Figure 2 shows the regret of different algorithms in the case without knowledge of expected rewards. We can see that in the non-boundary cases, UCB-ALP and UCB-EALP achieves regret that is very close to UCB-PB and outperforms UCB-FLP. Interestingly, we can even see that UCB-ALP achieves slightly lower regret than UCB-PB in the case with $\rho = 0.41$. This is because under UCB-PB, the better context may be skipped and wasted if it does not have the highest UCB. In contrast, the UCB-ALP algorithm may allocate certain resource to the better context, even when it does not have the highest UCB. On the boundary case, the regrets of UCB-ALP and UCB-EALP become larger than that of UCB-PB, but are still sublinear in $T$.

## G.2 Multi-Context Systems

Next, we study a multi-context scenario with $J = 10$ contexts, $K = 5$ arms, and Bernoulli rewards. Specifically, the context distribution vector is $\boldsymbol{\pi} =$

$[0.025, 0.05, 0.075, 0.15, 0.2, 0.2, 0.15, 0.075, 0.05, 0.025]$. The expected reward of action $k$ under context $j$ is $u_{j,k} = \frac{jk}{JK}$. One boundary in this system is $q_5 = 0.5$. We study the cases with average budget $\rho = 0.49, 0.5$, and $0.51$, respectively. In this case, it is difficult to calculate the expected total reward obtained by the oracle solution. Thus, we calculate the regret by comparing with the upper bound, i.e., $\widehat{U}(T, B) = Tv(\rho)$.

Figure 3: Comparison of algorithms for the multi-context systems with perfect knowledge ($Q_5 = 0.5$), (a) $\rho = 0.49$, (b) $\rho = 0.5$, (c) $\rho = 0.51$.

Figure 3 shows the regret of different algorithms in the case with known expected rewards. In the non-boundary cases, both the ALP and EALP algorithm achieve similar performance as in the two-context case. The regret of EALP is even lower than FLP in the boundary case, since the ratio of contexts that are executed with correct probability is higher than that in the two-context systems.

Figure 4: Comparison of algorithms for the multi-context systems without perfect knowledge ($Q_5 = 0.5$), (a) $\rho = 0.49$, (b) $\rho = 0.5$, (c) $\rho = 0.51$.

Figure 4 shows the regret of different algorithms in the case without knowledge of expected rewards. We can see that all algorithms achieve sublinear regret, but the difference between the non-boundary cases and the boundary case is small. This is rooted in the fact that when the number of contexts and the number of actions are large, it requires more time to learn the expected rewards. Hence, the constant in the $\log T$ term is much larger than that in the $\sqrt{T}$ term, and the $\log T$ term dominates the regret and the impact of the $\sqrt{T}$ term could be small. Exploring the structure of the reward function in contextual bandits, e.g., similarity [6] and linearity [5], to reduce the exploration time is part of our future work.