[Reviews · NeurIPS 2015]

Submitted by Assigned_Reviewer_1

Review after rebuttal : - I agree that it is not exactly the same, but it seems close enough to me : if the horizon is large enough, the coupling of the context vanishes (and given that, I don't understand why . So I am not sure why each occurrence of a context cannot be treated as a separate instance of a bandit problem). - To me, this paper would be much better with a clean (the supplementary material's bound has to be reworked) upper bound in term of arm gaps for the problem dependent case, and ideally matching lower bounds in problem dependent and independent cases.

----------------------

The authors consider the problem of a budgeted and time constrained contextual bandit. The main finding of their paper is that if the number of context and actions is finite, and if the time constraint T is sufficiently large (with respect to the problem at hand) then they prove that a regret of order \log(T) is achievable, which is a new result for contextual bandits.

The results seem thorough, and the writing is precise. The results are interesting but not very surprising. Indeed, since the number of contexts J and arms K are finite, this bandit problem can be seen as a a time constrained and budgeted bandit problem with JK arms, which is also action constrained since at each time, not all JK arms are accessible (only the K arms corresponding to the context). Since the number of arms is finite, for T large enough, the regret is \log(T) as in action constrained bandit problems. So I am not sure how innovative this paper is - maybe the authors can clarify this in their rebuttal.

I have a few questions regarding this paper : - In Theorem 2, the regret inn case 2) is O(\sqrt(T) + ...). But is it really O(\sqrt(T) + ...)? Shouldn't it be O(\sqrt(KT) + ...) at least? - Still in Theorem 2, the regret is either O(KJ \log(T)) or O(\sqrt(T) + KJ\log(T)) depending on the configuration of the arms. In general, problem dependent bounds are expressed in a more refined way with the sum of inverse of arm gaps. Isn't it possible to do something similar here? I understand that you have KJ instead of a certain sum of gaps because your proof is based on events characterizing the correct orders of the UCB, and not concentration bounds on their gaps. But is it an optimal idea? - It would be interesting to have lower bounds for this problem, ideally problem dependent and problem independent. Trivial lower bounds are the ones of classical bandit which correspond to cost equals 0 and context which is always the same. Can it be refined to take into account the complexity of the context?
Summary: I am still not convinced by the originality of this setting - and therefore remain neutral about this paper.

-------------------------

I have the impression that the setting of this paper is very close to a budgeted bandit problem with action constraint.

So I am not sure how innovative this paper is - maybe the authors can clarify this in their rebuttal.

Submitted by Assigned_Reviewer_2

(Light Review)

The paper considers contextual bandits with budget and time constraints. The analysis is restricted to finite discrete contexts, fixed costs and a single budget constraint. While the setting is simplified, these assumptions are still reasonable in practice. Importantly, it results in algorithms that achieve O(log T) regret by introducing the adaptive linear programming approximation of the oracle.

The paper is clear and well documented. Relevant work is cited and motivation is explicit. This is a solid theoretical paper. The practical aspects are only briefly discussed in Section 5 and 6 and no experimental results are provided. This being said, the contributions are significant: as far as I know, the authors are the first to obtain logarithmic regret in the case of constrained contextual bandits. The proofs are convincing (although I did not check them all in full detail).
Summary: This is a good paper, making appropriate simplified assumptions to obtain O(log T) regret algorithms. The practical aspects might be discussed into more detail.

Submitted by Assigned_Reviewer_3

This paper considers a contextual bandit problem with budget constraint B and known time horizon T. For each trial, the agent observes a context j chosen according to a fixed probability distribution \pi over a context set {1,..,J} and takes an action k from {0} \cup {1,..,K}, where 0 means "skip". If k = 0, then the agents gets no reward and incurs no cost. If k > 0, then the agent gets reward y and incurs cost c_{j,k}, where y is chosen according to a fixed probability distribution over [0,1] which depends on j and k, and c_{j,k} > 0 is a parameter known to the agent. The goal of the agent is to maximize the cumulative reward under the condition that the cumulative cost never exceeds the budget B.

The main body of the paper is devoted to a special case where \pi is known and c_{j,k} = 1 for all j and k, and the paper gives an algorithm whose regret bound is O(log T) under a mild condition. The paper extends the algorithm to the general case where \pi is unknown and c_{j,k}'s are arbitrary.

A more general setting is considered in [16], but it gives an algorithm with O(\sqrt T) regret. So the paper is possibly the first work with O(log T) regret, although the mild condition should be satisfied. The algorithm is based on an approximation algorithm for computing the optimal action sequence when all the statistics are known. The algorithm is simple but very interesting.

The paper is well motivated and very clearly written. It would be more interesting and significant if O(log T) regret always holds, otherwise \Omega(\sqrt T) lower bound is proved in the boundary cases.

All the results seems to be based on Lemma 1, but I'm not convinced of its correctness. I do not think the optimal action sequence always satisfies (2) for all trials, and so I do not understand why the average constraint is a relaxed version of the hard constraint. Please give a brief explanation. ==> Now I'm convinced of Lemma 1 by the rebuttal. Thanks

Minor comments: Is it NP-hard to exactly obtain the optimal action sequence? Can the results be generalized to the case where T is unknown? Can we assume u^*_1 > u^*_2 > ... > u^*_J

without loss of generality even for the unknown context distribution case?
Summary: I strongly recommend this paper for acceptance.

Submitted by Assigned_Reviewer_4

Since this is a short review, I only ask one question: what is the reason that boundary cases have to be analyzed differently? It doesn't seem that rho = q_j appears anyhow special, especially for j > 1.
Summary: This paper analyzes contextual bandits against a stronger baseline: one that uses adaptive linear programming to decide to pass on certain percent of suboptimal rounds (what the authors call budget). It has interesting setup and unique solutions.

Submitted by Assigned_Reviewer_5

This paper considers a fairly simple budget constrained cb problem with discrete contexts and actions. The main contribution is the computational efficiency of the algorithm (via LP approximation) and the log(T) regret bound it achieves.

This paper is very well written and easy to follow.

Suggestions: in future work, consider using a parametric model for the reward. This is useful when the number of contexts is large. In section 4.1, when combining UCB with ALP, be more clear about the rationale using in the UCB-ALP algorithm. For example, I think you are following the optimism in the face of uncertainty. It's probably equivalent to the policy update in UCRL2. It'll be great to include some numerical experiments in extended version of the paper.

you may want to this paper: http://arxiv.org/abs/1506.03374
Summary: This paper merits acceptance since it provides an efficient algorithm for constrained cb with optimal regret in most cases. It is interesting and indeed an improvement over existing work.

Author Feedback
Author rebuttal: Reviewer 1:
- It is not obvious to us that treating context-arm pairs as arms would lead to a logarithmic regret. There are two sources of difficulty here: the available "arms" at each time are random, and the contexts are coupled over time due to the budget constraint. With limited budget, viewing each occurrence of a context as a separate instance of a bandit problem, claiming a logarithmic regret for each such instance, and combining the results to get a logarithmic regret is not possible. In fact, without a proper tradeoff between the immediate and future rewards, the agent may suffer from higher regret. Our work proposes an approximate oracle for dynamically allocating budget over contexts, which enables us to combine it with the UCB method to achieve an O(\log(T)) regret.
- Minor issues: a) It is O(\sqrt{T} + ...). This term comes from the approximate oracle and its specific representation is (u_1^* - u_J^*)\sqrt{\rho(1-\rho)}\sqrt{T}. b) The bounds depend on the gaps, but are much more complex. Specific representations are given in the supplementary material. c) We can show an O(log T) lower bound. It is still an open problem whether O(\sqrt T) is a lower bound for the boundary cases with J > 2.

Reviewer 2:
- It is true that the optimal sequence may not satisfy (2) in exactly the same format for some trials. However, Lemma 1 still holds because any feasible solution satisfies the hard constraint for all trials, and we can show that the policy satisfies the average constraint by taking expectation over all trials. Specifically, let T_j be the number of pulls under context j for any trial under any feasible policy with known statistics. Let p_j = E[T_j]/(\pi_j T), which satisfies 0 <= p_j <= 1. Then the expected reward becomes \sum_j E[T_j] u_j^* = T\sum_j p_j \pi_j u_j^*. Further, because the hard budget constraint is met for all trials, i.e., \sum_j T_j <= B, we have \sum_j p_j \pi_j = \sum_j E[T_j]/T <= B/T. This proof will be added to the revision.
- Minor issues: a) The general case is NP-hard because it can be mapped to a knapsack problem. b) We need to know the normalized remaining budget, b/\tau. c) This assumption can still be made WLOG because we only need the order of the rewards.

Reviewer 3:
- We will discuss in more detail about the intuition behind UCB-ALP in the revision, provide more simulation results in the extension (some preliminary results are in Appendix F of the supplementary material), and consider parametric rewards in future work.
- Thanks for bringing this recent paper, which was not cited in our submission because of its unavailability before the NIPS deadline, to our attention. The paper considers general reward and constraints. However, it focuses on a finite policy set as [16]. We will cite this paper in our work.

Reviewer 4:
- More detailed discussions on practical issues will be provided. Some simulation results can be found in the supplementary material.

Reviewer 6:
- For non-boundary cases, the margin between \rho and \q_j tolerates the fluctuation of b_\tau/\tau under UCB-ALP. When T is large, b_\tau/\tau will not cross boundaries with high probability, and the errors can be compensated later. However, in boundary cases, b_\tau/\tau fluctuates near the boundary and the boundary-crossing probability does not diminish even for large T, which causes an O(\sqrt T) regret. We are not sure if there are other algorithms with logarithmic regret for these cases.

Reviewer 7:
- We have completed the study for cases with unknown context distribution and heterogeneous costs. They are available in the supplementary material. We have shown that modified versions of ALP and UCB-ALP can achieve similar performance as that in the simplified case. The design and analysis are more challenging. For example, we need to deal with the temporal dependency in the unknown context distribution case, and the intra-context coupling in the heterogeneous-cost case. Due to space limitations, we only discussed the differences and the key ideas in the main body, and put the detailed results in the supplementary material (Appendix D & Appendix E).
- We agree that a contextual bandit problem can be viewed as a special case of the reinforcement learning problem in [A.1,A.2]. However, typical algorithms such as NCRL [A.1] and NCRL2 [A.2] do not consider budget constraints. It is not clear to us how to extend these algorithms to constrained contextual bandits. We will discuss the relationship and differences between the RL problems and our constrained contextual bandits.
- We compare with FLP and FLP-UCB because the previous work [16] and [19] use fixed budget constraints for a single step decision. More discussions about the reason of choosing these benchmarks will be provided.

[A.1] P. Auer, et al. "Logarithmic online regret bounds for undiscounted reinforcement learning." NIPS'06.
[A.2] P. Auer, et al. "Near-optimal regret bounds for reinforcement learning." NIPS'08.